# Steering and Rectifying Latent Representation Manifolds in Frozen Multi-modal LLMs for Video Anomaly Detection

**Zhaolin Cai**[1], **Fan Li**[1,2*], **Huiyu Duan**[3*], **Lijun He**[2], **Guangtao Zhai**[3,4*]
[1]Xinjiang University    [2]Xi'an Jiao Tong University
[3]Shanghai Jiao Tong University    [4]Shanghai AI Laboratory

## Abstract

Video anomaly detection (VAD) aims to identify abnormal events in videos. Traditional VAD methods generally suffer from the high costs of labeled data and full training, thus some recent works have explored leveraging frozen multi-modal large language models (MLLMs) in a tuning-free manner to perform VAD. However, their performance is limited as they directly inherit pre-training biases and cannot adapt internal representations to specific video contexts, leading to difficulties in handling subtle or ambiguous anomalies. To address these limitations, we propose a novel intervention framework, termed **SteerVAD**, which advances MLLM-based VAD by shifting from passively reading to actively steering and rectifying internal representations. Our approach first leverages the gradient-free representational separability analysis (RSA) to identify top attention heads as latent anomaly experts (LAEs) which are most discriminative for VAD. Then a hierarchical meta-controller (HMC) generates dynamic rectification signals by jointly conditioning on global context and these LAE outputs. The signals execute targeted, anisotropic scaling directly upon the LAE representation manifolds, amplifying anomaly-relevant dimensions while suppressing inherent biases. Extensive experiments on mainstream benchmarks demonstrate our method achieves state-of-the-art performance among tuning-free approaches requiring only 1% of training data, establishing it as a powerful new direction for video anomaly detection. The code is available at `https://github.com/CebCai/SteerVAD`.

## 1 Introduction

Video anomaly detection (VAD) aims to identify events that deviate from expected patterns, which plays a critical role in intelligent surveillance (Sultani et al., 2018), industrial quality control (Roth et al., 2022; Liu et al., 2024b), and autonomous systems (Yao et al., 2023; Bogdoll et al., 2022). While traditional paradigms, encompassing supervised (Liu et al., 2018; Landi et al., 2019), weakly supervised (Wu et al., 2024b) and unsupervised approaches (Lv et al., 2021; Liu et al., 2021), have demonstrated notable success but they are constrained by the reliance on large-scale training data. This dependency not only incurs substantial computational and annotation costs but also limits their ability to generalize to unseen scenarios, hindering real-world deployment (Noghre et al., 2025).

The advent of multi-modal large language models (MLLMs) (Zhu et al., 2023; Liu et al., 2023) has opened a new frontier for VAD by leveraging their powerful capabilities. While fine-tuning these models for VAD is effective (Zhang et al., 2024a;b), these methods reintroduce the burden of computational costs and data requirements, undermining the advantages of pre-trained models. Recent studies have shifted towards tuning-free paradigms (Shao et al., 2025), which utilize frozen MLLMs to detect anomalies by their generated interpreting text or features (Zanella et al., 2024; Ye et al., 2025). Despite their effectiveness, these approaches are constrained by their passive nature. They overlook two critical deficiencies deeply rooted within the models. (1) The first is inherent representational bias of MLLMs, since they are pre-trained on web-scale corpora, developing a feature space optimized for frequent and prototypical concepts. As a result, their representations exhibit reduced sensitivity to subtle and rare patterns typical in anomalous events, leading to missed or biased detection. (2) The second is contextual ambiguity. Since the semantic meaning of a local action is determined by its global context, passively relying on isolated features can lead the model

---

*Corresponding authors.

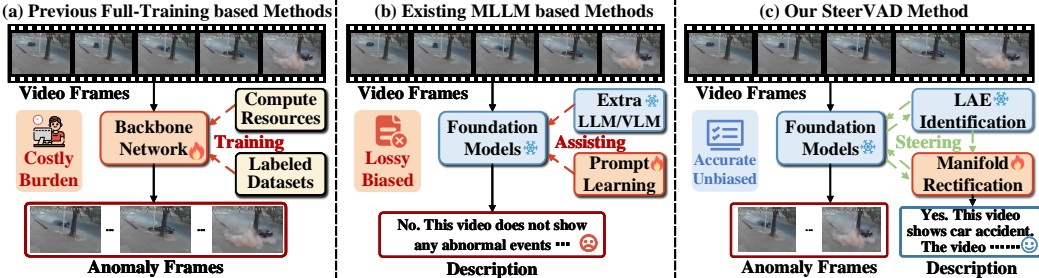

**Figure 1:** Comparison of traditional full-training methods, existing tuning-free methods and our proposed SteerVAD. Our method overcomes the issue of costly training resources and inherent biases with minimal data required from pre-trained foundation models compared to previous methods.

to produce confounding representations for visually similar but contextually distinct events. These issues are not mere surface-level classification errors, but stem from structural flaws in the internal representations of MLLMs, highlighting the inherent limitations of passive interpretation.

To address these limitations, we advocate a perspective shift from passive feature reading to active geometric intervention. Our approach builds on the well-established manifold hypothesis (Whiteley et al., 2022), which posits that high-dimensional natural data concentrates on low-dimensional structures. We extend this hypothesis to the latent geometry of MLLMs. Within their high-dimensional feature space, the representations for a semantic class coalesce into a coherent, low-dimensional structure, which we term representation manifold. From this geometric perspective, the identified deficiencies can be re-contextualized. *Inherent representational bias* manifests as a systemic proximity between the manifolds of normal and anomalous events. The feature space optimized for frequent normal patterns fails to allocate sufficient geometric separation for their low-frequency abnormal counterparts. This proximity is compounded by *contextual ambiguity*, manifesting as the local geometric entanglement of these manifolds. In these entangled regions, representations of visually similar but semantically distinct events become indistinguishable, making robust separation extremely challenging. This reframing highlights that the core challenge is not merely the classification of static features, but the dynamic steering and rectification of the underlying latent manifolds.

In this paper, we introduce SteerVAD, a novel tuning-free framework that implements this geometric intervention by steering latent representation manifolds within a frozen MLLM. SteerVAD operationalizes our geometric insight through two synergistic innovations. First, we develop a gradient-free representational separability analysis (RSA) to identify internal attention heads termed latent anomaly experts (LAEs), whose representations are effective for separating normal and anomalous events, exhibiting strong potential for rectification. This identification allows our intervention to be focused and efficient, avoiding unnecessary manipulation of the entire model. Second, we design a lightweight hierarchical meta-controller (HMC) that learns to execute dynamic, context-dependent geometric transformations. Conditioned on a global understanding of the scene, the HMC executes targeted anisotropic scaling to the features produced by identified LAEs, thereby actively reshaping their corresponding manifolds by amplifying anomaly-relevant dimensions while suppressing biased ones. By calibrating on only 1% of the training set, this mechanism effectively disentangling the representations of normal and anomalous events without any fine-tuning on the pre-trained model.

Our main contributions are as follows:

- We introduce a novel intervention paradigm for tuning-free video anomaly detection that moves beyond passive feature interpretation to active geometric intervention. Our framework is the first to operationalize this paradigm by directly steering and reshaping latent representation manifolds within completely frozen MLLM.

- We introduce representational separability analysis (RSA), a novel gradient-free geometric method to precisely identify latent anomaly experts whose internal feature spaces are most aligned for VAD, ensuring the data-efficiency of our framework.

- We design a hierarchical meta-controller (HMC) that generates context-aware signals to perform anisotropic manifold scaling, dynamically disentangling class representations to overcome pre-training biases and contextual ambiguities.

- We establish new state-of-the-art performance on UCF-Crime and XD-Violence datasets among tuning-free methods using a frozen MLLM, demonstrating that targeted, data-efficient intervention method as a promising way against expensive fine-tuning approaches.

## 2 RELATED WORK

**Traditional Video Anomaly Detection.** Research in video anomaly detection (VAD) has traditionally focused on models trained specifically for the task, categorized by the level of supervision. Supervised methods (Liu et al., 2018; Landi et al., 2019) have shown strong performance by training on extensive datasets with frame-level labels, but the high cost of annotation limits their practical application. To mitigate this, weakly-supervised approaches (Li et al., 2022a; Wang et al., 2024) leverage more accessible video-level labels, often using multiple instance learning (MIL) frameworks. Unsupervised methods (Tur et al., 2023; Zaheer et al., 2022; Thakare et al., 2023) are trained exclusively on normal data. These models learn to identify anomalies as deviations from a learned normality model, typically based on reconstruction error (Xu et al., 2017) or future frame prediction (Luo et al., 2022). While effective, these traditional paradigms require heavily training and struggle to generalize to unseen anomalies, motivating a shift towards new paradigms.

**Video Anomaly Detection with LLMs and MLLMs.** The advent of large language models (Vaswani et al., 2017; Touvron et al., 2023) and multi-modal large language models (MLLMs) (Zhu et al., 2023; Li et al., 2023; Wu et al., 2024d), has created a new way for VAD. Initial efforts involves adapting these models through supervised fine-tuning (Zhang et al., 2023; Yuan et al., 2024). While effective, these methods reintroduce the need for large labeled datasets and high computational costs. A more recent and promising direction is the tuning-free paradigm, which uses MLLMs in a zero-shot or few-shot capacity (Shao et al., 2025). These methods involve querying the MLLM with modified inputs to generate enhanced textual descriptions or decisions about video, which are then parsed to identify anomalies (Zanella et al., 2024; Ye et al., 2025). Although these approaches effectively leverage the MLLMs without expensive training, they treat the representations of the model as static and immutable, which makes them susceptible to pre-training biases and ambiguity, leaving a crucial gap in how to actively utilize the powerful internals of these models.

**Internal Analysis of LLMs and MLLMs.** Recent research in mechanistic interpretability of LLMs and MLLMs provides the foundation for intervention (Luo & Specia, 2024). Studies have shown that complex behaviors in foundation models can be localized to specific modules, such as individual attention heads or neuron groups (Zheng et al., 2025), which act as functional circuits (Zhou et al., 2024; Bi et al., 2024). Building on this insight, the field of model editing has the ability to make targeted modifications to model weights to alter specific behaviors (Wang & Veitch, 2025; Zweiger et al., 2025). These techniques have been applied to language tasks or involve static, input-independent interventions (Nguyen et al., 2025; Mahmoud et al., 2025). The application of dynamic, context-dependent rectification of internal geometric structures for a complex, spatio-temporal reasoning task remains an unexplored frontier. Our work is the first to bridge this gap by introducing actively geometric rectifications on the representation manifolds with frozen MLLM for VAD.

## 3 METHODOLOGY

### 3.1 PRELIMINARY: REPRESENTATION MANIFOLDS IN MLLMS

Our theoretical approach is based on the manifold hypothesis (Whiteley et al., 2022; Genovese et al., 2010; Boissonnat & Ghosh, 2010), which posits that high-dimensional data concentrates on or near a low-dimensional manifold. We extend this to the latent space of MLLMs, where an attention head $h : \mathcal{V} \rightarrow \mathbb{R}^{d_{\text{head}}}$ acts as a feature extractor that maps semantically similar video inputs into geometrically proximal representations, causing the varied manifestations of a class to coalesce into a coherent structure. We model these emergent structures as representation manifolds. Consequently, for a given head $h$, all normal event representations form a manifold $\mathcal{M}_{\text{norm}}$, while anomalous events form a manifold $\mathcal{M}_{\text{anom}}$, both embedded within the head's high-dimensional feature space. A rigorous topological analysis of these structures is detailed in Appendix A.

Visualizations using UMAP, Figure 2 provides an intuitive illustration consistent with this geometric perspective. The features appear to form two distinct yet intertwined manifolds, visually confirming the problems of representational bias and contextual ambiguity. This geometric entanglement makes reliable separation by a simple classifier difficult. Our approach moves beyond passively accepting this topology, we aim to learn a minimal, context-dependent geometric transformation $T_{\mathbf{c}}$ that actively rectifies these manifolds to achieve clear differentiation.

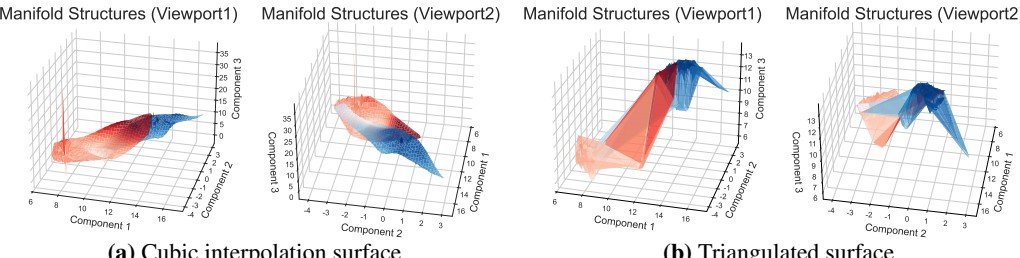

**(a)** Cubic interpolation surface        **(b)** Triangulated surface

**Figure 2:** 3D UMAP visualization of representation manifolds of normal (blue) and anomalous (red) events from InternVL, illustrating their geometric structure. Each manifold is rendered from two perspectives using (a) cubic interpolation and (b) triangulation.

## 3.2 FRAMEWORK OVERVIEW

As shown in Figure 3, SteerVAD enables geometric intervention within a frozen MLLM for VAD. The framework rests on two principles: identification of relevant representational subspaces and rectification of them. To accomplish the first, a representational separability analysis (RSA) identifies few internal attention heads that inherently aligned with VAD, which we term latent anomaly experts (LAEs). Then the hierarchical meta-controller (HMC) drives rectification by integrating the global context with LAE outputs to generate steering signals. These signals perform geometric transformations on LAE feature manifolds, amplifying anomaly cues and suppressing biases. The rectified representations are aggregated and scored by anomaly scorer to produce final decision and curves. Detailed component implementation and analysis of SteerVAD are provided in Appendix B.

**Problem Formulation**    During the inference, given an input video stream $\mathcal{V}$, the stream is first partitioned into a sequence of non-overlapping temporal segments, $\mathcal{V} = \{S_1, S_2, \ldots, S_T\}$. For each segment $S_t$, our objective is to compute an anomaly probability $p_t \in [0, 1]$ in a real-time, single-pass manner. The identification of LAEs and the training of the HMC and scorer are performed in a preceding offline calibration phase, utilizing a small, representative labeled dataset $\mathcal{D}_{\text{calib}}$.

## 3.3 REPRESENTATIONAL SEPARABILITY ANALYSIS

To identify sub-modules within frozen MLLM that are inherently aligned with the video anomaly detection (VAD) task, we propose representational separability analysis (RSA) to discover latent anomaly experts (LAEs), defined as specific attention heads whose feature representations exhibit a high degree of geometric separability between normal and anomalous patterns.

**Separability Metric.**    We compute the Inter-to-Intra Scatter Ratio as the RSA score to quantify geometric suitability of each head. This metric measures the ratio of between-class separation to within-class compactness. For head $h_{l,k}$ (layer $l$, index $k$), the RSA score is computed as:

$$S_{\text{RSA}}(l, k) = \frac{\|\boldsymbol{\mu}_{\text{anom}}^{(l,k)} - \boldsymbol{\mu}_{\text{norm}}^{(l,k)}\|_2^2}{\sigma_{\text{anom}}^2(l, k) + \sigma_{\text{norm}}^2(l, k)} \tag{1}$$

where $\boldsymbol{\mu}_{\text{anom/norm}}^{(l,k)}$ are centroids for anomalous and normal samples respectively, and $\sigma_{\text{anom/norm}}^2(l, k)$ are their corresponding intra-cluster variances. A high $S_{\text{RSA}}$ score signifies that a head maps normal and anomalous inputs to distinct and compact clusters, making them aligned with VAD task.

**Feature Extraction and LAE Selection.**    The LAE selection is performed on the calibration set $\mathcal{D}_{\text{calib}}$ via a single-forward pass for each video through the frozen MLLM. We extract and store features from all attention heads to build feature banks, compute the $S_{\text{RSA}}$ score and select the top-$K$ heads with the highest scores as LAEs. This process efficiently pinpoints the most informative internal subspaces of the MLLM, providing optimal targets for dynamic rectification.

## 3.4 HIERARCHICAL META-CONTROLLER FOR DYNAMIC MANIFOLD RECTIFICATION

The hierarchical meta-controller (HMC) dynamically orchestrates manifold rectification via a single MLLM forward pass. The HMC receives two concurrent information streams extracted from the frozen MLLM: the fine-grained feature vectors $\{\mathbf{h}_i\}_{i=1}^K$ from the $K$ LAEs, and a global context vector $\mathbf{c} \in \mathbb{R}^{d_{\text{model}}}$. The context vector $\mathbf{c}$ is obtained by extracting the final hidden state corresponding to the first generated token, which serves as a holistic semantic summary of the scene by the MLLM. The hierarchical design of the HMC decouples the rectification task into two synergistic levels.

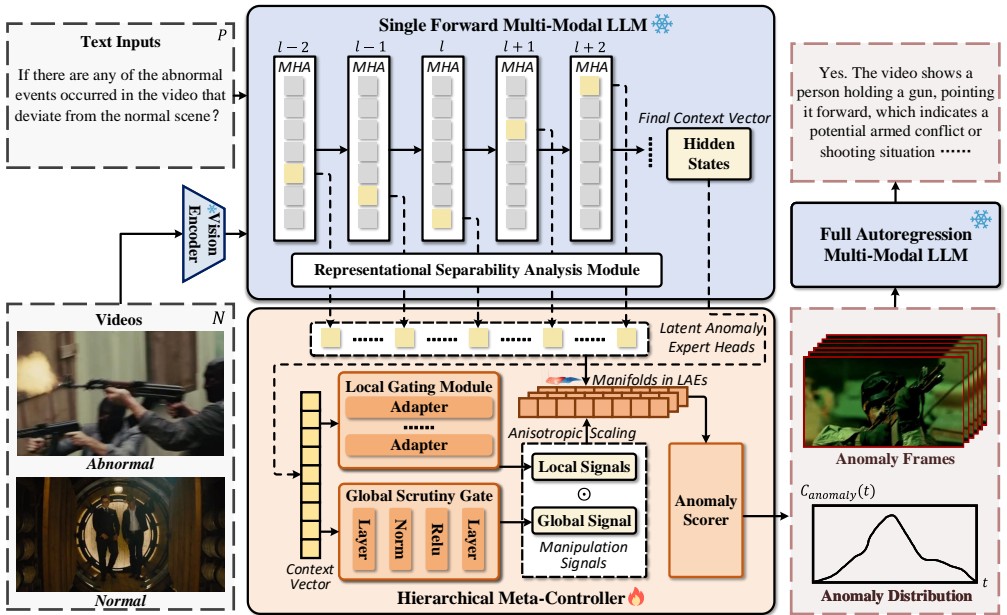

**Figure 3:** Framework overview of SteerVAD. We first apply the Representational Separability Analysis to find top $K$ Latent Anomaly Experts inside frozen MLLM. During the single pass, The global context vector $\mathbf{c}$ and LAE features $\{\mathbf{h}_i\}$ are extracted. The Hierarchical Meta-Controller ingests these signals, using Global Scrutiny Gate and Local Gating Module to generate manipulation signals ($s_{\text{global}}$, $\{\mathbf{g}_i\}$). These signals perform Anisotropic Manifold Scaling to rectify LAE features. A lightweight Anomaly Scorer receives the rectified features and outputs the final anomaly curve. Detected anomalous frames can be passed to the full MLLM to produce a textual explanation.

**Global Scrutiny Gate.** The global scrutiny gate (GSG) computes a holistic suspicion score $s_{\text{global}}$ as a global signal to quantify the overall possibility of an anomaly. It is implemented as a small multi-layer perceptron (MLP) that takes the global context vector $\mathbf{c}$ as input:

$$s_{\text{global}} = (\sigma \circ \text{Linear}_2 \circ \text{ReLU} \circ \text{BN} \circ \text{Linear}_1)(\mathbf{c}) \in [0, 1] \tag{2}$$

where $\text{Linear}_1 \in \mathbb{R}^{d_{\text{model}} \times d_{\text{hidden}}}$, $\text{Linear}_2 \in \mathbb{R}^{d_{\text{hidden}} \times 1}$, BN denotes `BatchNorm1d`, and $\sigma$ is the sigmoid function. A value near 0 indicates a benign scene, prompting the HMC to remain quiescent, while a value near 1 signals a high-suspicion event that requires significant representational steering. This scalar acts as a master switch, determining the overall intensity of the subsequent rectification.

**Local Gating Module.** While the global gate provides a coarse-level signal, the local gating module generates a fine-grained, per-feature control signal for each LAE. It consists of $K$ parallel, lightweight adapter networks. Each adapter is a low-rank network that takes the global context $\mathbf{c}$ as input and outputs a unique, dense steering vector $\mathbf{g}_i \in \mathbb{R}^{d_{\text{head}}}$:

$$\mathbf{g}_i = \tanh(\text{Linear}_{\text{up}}(\text{Linear}_{\text{down}}(\mathbf{c}))) \tag{3}$$

where $\text{Linear}_{\text{down}} \in \mathbb{R}^{d_{\text{model}} \times r}$ projects the context vector into a low-rank bottleneck space, and $\text{Linear}_{\text{up}} \in \mathbb{R}^{r \times d_{\text{head}}}$ projects it back to the feature dimension of the expert head. The rank $r$ is a hyperparameter set to a small value to minimize trainable parameters. The hyperbolic tangent `tanh` activation constrains the output local signals $g_i$ to $[-1, 1]$, enabling the controller to learn not only to amplify feature dimensions in positive values but also to actively suppress them.

**Anisotropic Manifold Scaling.** The control signals generated by the GSG and LGM converge at the core rectification mechanism, anisotropic manifold scaling. Our goal is to apply a targeted, context-dependent geometric transformation to the LAE feature space. We operationalize this transformation through a simple yet powerful element-wise operation, ensuring it incurs negligible computational overhead. A detailed geometric interpretation of this operation is provided in the Appendix A. For each LAE feature $\mathbf{h}_i$, the rectified feature $\mathbf{h}_i'$ is computed as:

$$\mathbf{h}'_i = \mathbf{h}_i \odot (1 + s_{\text{global}} \cdot \mathbf{g}_i) \tag{4}$$

This formulation instantiates the geometric transformation as a stable, residual modulation. The global signal $s_{\text{global}}$ governs the intensity of transformation, while the local vector $\mathbf{g}_i$ provides its critical anisotropy, enabling the targeted stretching and compressing of the manifold along specific semantic axes potentially linked to pre-training biases.

## 3.5 ANOMALY AGGREGATION AND SCORING

The anomaly scorer processes the rectified features from selected LAEs by estimating frame level anomaly probabilities and then applying temporal smoothing to yield final coherent anomaly curves.

**Probabilistic Anomaly Scoring.** For each segment $S_t$, the rectified feature vectors $\{\mathbf{h}'_i \in \mathbb{R}^{d_{\text{head}}}\}_{i=1}^{K}$ from selected LAEs are aggregated via concatenation into a single vector $F_{\text{final}}^{(t)} = \text{Concat}(\mathbf{h}'_1, \ldots, \mathbf{h}'_K) \in \mathbb{R}^{K \cdot d_{\text{head}}}$. The aggregated vector is then input to anomaly scorer, a logistic regression classifier that maps the feature to a frame-level anomaly probability $p_t$:

$$p_t = \sigma(\mathbf{w}^T F_{\text{final}}^{(t)} + b) \tag{5}$$

where $\mathbf{w}$ and $b$ are the learnable weights and bias, and $\sigma$ is the sigmoid function. This scorer ensures minimal computational overhead and prevents overfitting on the small calibration dataset, thereby focusing the learning complexity within the hierarchical meta-controller.

**Anomaly Curve Generation.** To mitigate noise from transient visual fluctuations in the raw anomaly probability sequence $\{p_1, p_2, \ldots, p_T\}$, we employ temporal smoothing via 1D Gaussian convolution. This process, predicated on the temporal locality of anomalous events, yields a stable anomaly curve $A_t$ by evaluating each point within its temporal neighborhood. The final score $A_t$ is computed as a weighted average over a temporal window:

$$A_t = \frac{\sum_{j=1}^{T} p_j \cdot G(t - j; \sigma_g)}{\sum_{j=1}^{T} G(t - j; \sigma_g)} \tag{6}$$

where $G(x; \sigma_g) = \exp(-x^2/(2\sigma_g^2))$ is the Gaussian kernel with standard deviation $\sigma_g$. and normalization ensures unit weight sum. This filtering suppresses transient noise while preserving sustained anomalies, yielding a smoother, more reliable detection signal for practical deployment.

**Post-Hoc Explainability.** When segments $S_t$ are flagged as anomalous, the corresponding video frames can be concatenated and re-submitted to the frozen auto-regressive MLLM. The model then generates a textual description, offering explanations for the anomaly alert and enhancing the transparency and trustworthiness of our framework.

## 3.6 TRAINING OBJECTIVE

The trainable parameters of our framework consist of the HMC including GSG, LGM and anomaly scorer. We train the framework with a composite objective that jointly maximizes classification accuracy and enforces an inductive bias for anomaly detection. The main component of this objective is the binary cross-entropy (BCE) loss between the predicted anomaly probability $p_t$ and the ground-truth $y \in \{0, 1\}$:

$$\mathcal{L}_{\text{BCE}} = -[y \log(p_t) + (1 - y) \log(1 - p_t)] \tag{7}$$

To mitigate false positives by preventing the model from overreacting to benign activity, we introduce a sparsity-inducing regularization on the global signal $s_{\text{global}}$ for normal samples. Let $\mathcal{B}_{\text{norm}}$ be the set of normal samples in a training batch. The regularization loss is:

$$\mathcal{L}_{\text{reg}} = \frac{1}{|\mathcal{B}_{\text{norm}}|} \sum_{j \in \mathcal{B}_{\text{norm}}} (s_{\text{global}}^{(j)})^2 \tag{8}$$

where $s_{\text{global}}^{(j)}$ is the global signal for the $j$-th normal sample. This L2 penalty encourages the controller to remain dormant ($s_{\text{global}} \approx 0$) for normal inputs.

The final objective is a weighted combination of these two losses:

$$\mathcal{L}_{\text{total}} = \mathcal{L}_{\text{BCE}} + \lambda_{\text{reg}}\mathcal{L}_{\text{reg}} \qquad (9)$$

where $\lambda_{\text{reg}}$ controls the regularization strength. This composite objective trains the HMC to activate its feature-modulating capacity only when necessary. Promoting sparsity in the activation of the global gate enhances the robustness of the model and reduces the potential for false positives.

## 4 EXPERIMENTS

### 4.1 EXPERIMENTAL SETTINGS

**Datasets.** We evaluate our framework on two widely adopted, large-scale benchmarks for VAD.

- **UCF-Crime** (Sultani et al., 2018) contains 1900 untrimmed real-world surveillance videos consist of 13 distinct anomaly types, totaling over 128 hours. The official split contains 1610 videos (810 abnormal and 800 normal) for training and 290 (140 abnormal and 150 normal) for testing.
- **XD-Violence** (Wu et al., 2020) comprises 4754 untrimmed videos from movies, sports, and online sources. It includes 6 different anomaly categories. The dataset is divided into 3954 training videos and 800 test videos (500 abnormal and 300 normal).

We use a small, randomly sampled 1% subset of the training sets from each dataset for our LAE discovery and the training of the HMC. The full official test sets are used for evaluation.

**Evaluation Metric.** We adopt the standard evaluation metrics for each dataset to ensure a fair comparison with prior works. For UCF-Crime, we report the frame-level Area Under the Receiver Operating Characteristic Curve (AUC). This metric provides a comprehensive measure of the ability of a model to distinguish between normal and anomalous frames across all decision thresholds. For XD-Violence, we report the Average Precision (AP), which evaluates the precision-recall trade-off and is sensitive to the correct classification of positive (anomalous) instances. For both metrics, higher values indicate superior performance.

**Implementation Details.** SteerVAD is built upon the InternVL3 (Zhu et al., 2025) as the frozen MLLM backbone. For video processing, each video is divided into non-overlapping segments at an interval of 48 frames and uniformly sample $F = 16$ frames from each segment as the input to MLLM. For the offline calibration (LAE discovery and HMC training), we select the top $K = 4$ attention heads as the LAEs based on the RSA scores. The HMC and anomaly scorer are trained for 1000 epochs using the Adam optimizer with a learning rate of $1 \times 10^{-3}$ and a batch size of 64, the regularization weight $\lambda_{\text{reg}} = 0.1$, and the standard deviation for the Gaussian smoothing kernel $\sigma_g = 6$. All experiments were conducted on a single NVIDIA RTX A6000 GPU.

### 4.2 COMPARISON WITH STATE-OF-THE-ART METHODS

As detailed in Table 1, our proposed SteerVAD establishes a new state-of-the-art (SOTA) in the tuning-free video anomaly detection approaches, demonstrating consistent superiority on both the UCF-Crime and XD-Violence benchmarks. This strong performance is attributed to our active intervention mechanism, where a hierarchical meta-controller executes targeted geometric steering within the feature space. Unlike passive methods that rely on fixed representations from a pre-trained MLLM, our approach dynamically rectifies the feature space to counteract inherent model biases. This process selectively amplifies anomalous signals that would otherwise be obscured.

Furthermore, SteerVAD narrows the performance gap with fully-trained fine-tuned methods while operating with exceptional data efficiency. On UCF-Crime, our model achieves an AUC of 87.15%, which is highly competitive with the 89.51% AUC of the Holmes-VAD. Crucially, SteerVAD utilizing a minuscule fraction of the data demanded by its counterparts. This validates that leveraging the global understanding of an MLLM through targeted geometric steering presents a compelling and efficient alternative to expensive fine-tuning for adapting MLLMs to specialized downstream tasks.

### 4.3 VISUALIZATION OF LATENT SPACE RECTIFICATION

To provide a qualitative assessment of our manifold rectification strategy, we employ t-SNE to visualize the latent feature distributions from the XD-Violence and UCF-Crime datasets in Figure 4. The

**Table 1:** Comparison with existing methods on the UCF-Crime and XD-Violence datasets.

| Mode | Methods | Backbone | Params | UCF AUC (%) | XD AP (%) |
|------|---------|----------|--------|-------------|-----------|
| Weakly Supervised | Wu et al. (Wu et al., 2020) | I3D | - | 82.44 | 73.20 |
| | MIST (Feng et al., 2021) | I3D | - | 82.30 | - |
| | RTFM (Tian et al., 2021) | I3D | - | 84.30 | 77.81 |
| | S3R (Wu et al., 2022) | I3D | - | 85.99 | 80.26 |
| | MSL (Li et al., 2022b) | I3D | - | 85.30 | 78.28 |
| | UR-DMU (Zhou et al., 2023) | I3D | - | 86.97 | 81.66 |
| | MFGN (Chen et al., 2023) | I3D | - | 86.98 | 79.19 |
| | Wu et al. (Wu et al., 2024a) | ViT | - | 86.40 | 66.53 |
| | CLIP-TSA (Joo et al., 2023) | ViT | - | 87.58 | 82.19 |
| | Yang et al. (Yang et al., 2024) | ViT | - | 87.79 | 83.68 |
| | VadCLIP (Wu et al., 2024c) | ViT | - | 88.02 | 84.51 |
| Self Supervised | TUR et al. (Tur et al., 2023) | Resnet | - | 66.85 | - |
| | BODS (Wang & Cherian, 2019) | I3D | - | 68.26 | - |
| | GODS (Wang & Cherian, 2019) | I3D | - | 70.46 | - |
| Unsupervised | GCL (Zaheer et al., 2022) | ResNext | - | 71.04 | - |
| | DYANNET (Thakare et al., 2023) | I3D | - | 84.50 | - |
| Fine-Tuned | Holmes-VAU (Zhang et al., 2024b) | ViT+LLM | ∼2B | 88.96 | 87.68 |
| | Holmes-VAD (Zhang et al., 2024a) | ViT+LLM | ∼7B | 89.51 | 90.67 |
| Tuning-Free Multimodal VAD | *Full-Training Free* | | | | |
| | Zero-Shot CLIP (Radford et al., 2021) | ViT | ∼0.4B | 53.16 | 17.83 |
| | ZS ImageBind (Girdhar et al., 2023) | ViT | ∼1.2B | 55.78 | 25.36 |
| | LLAVA-1.5 (Liu et al., 2024a) | ViT+LLM | ∼7B | 72.84 | 50.26 |
| | LAVAD (Zanella et al., 2024) | BLIP-2+LLM | ∼14B | 80.28 | 62.01 |
| | EventVAD (Shao et al., 2025) | ViT+LLM | ∼7B | 82.03 | 64.04 |
| | *Few-Parameter Learning* | | | | |
| | VERA (Ye et al., 2025) | ViT+LLM | ∼8B | 86.55 | 70.54 |
| | HiProbeVAD (Cai et al., 2025) | ViT+LLM | ∼8B | 86.72 | 82.15 |
| | **SteerVAD (Ours)** | **ViT+LLM** | **∼8B** | **87.15** | **83.02** |

**Figure 4:** Visualization of manifold rectification via t-SNE on aggregated features from the LAEs for the XD-Violence and UCF-Crime datasets.

original features extracted by the frozen MLLM exhibit a high degree of overlap, where the representations of normal (blue) and anomalous (red) samples are largely indistinguishable. This observation is consistent across both datasets, highlights the representational bias that hinders downstream separability. In contrast, after the application of our hierarchical meta-controller, the rectified feature space is fundamentally restructured. The same samples now form two distinct, compact clusters with a large inter-class margin. This enhancement in separability qualitatively demonstrates that our method effectively remaps the latent manifold to mitigate the inherent bias, yielding representations more conducive to classification. Further visualizations are available in Appendix G.

## 4.4 ABLATION STUDIES

**Effectiveness of the Hierarchical Meta-Controller.** We validate the design of our hierarchical meta-controller (HMC) through the ablation studies presented in Table 2 (Part A). The strategy that trains a linear classifier on frozen raw LAE features, achieves only 81.33% AUC, confirming the need for feature rectification. We then explore context-agnostic modulation strategies. A static steer-

**Table 2:** Ablation studies on the UCF-Crime dataset, analyzing the impact of key components and expert selection strategies.

| Configuration / Strategy | GSG | LGM | Steering | AUC (%) |
|---|---|---|---|---|
| *Part A: HMC Component Ablation* | | | | |
| **Full Model** | ✓ | ✓ | **Anisotropic** | **87.15** |
| – w/o Global Gate | ✗ | ✓ | Anisotropic | 85.94 |
| – w/ Additive Steering | ✓ | ✓ | Additive | 85.02 |
| – w/o LGM (Static) | ✓ | ✗ | Static Scaling | 84.21 |
| – Linear Probing | ✗ | ✗ | NA | 81.33 |
| *Part B: Expert Selection Strategy Ablation* | | | | |
| **RSA (Ours)** | | | – | **87.15** |
| Mid-to-Last Layer | | | – | 84.68 |
| Random Selection | | | – | 69.57 |
| First-to-Mid Layer | | | – | 63.16 |

**Table 3:** Stability analysis of LAE selection and downstream performance across 10 independent runs with different random seeds on the 1% calibration subset.

| Seed ID | 7270 | 860 | 5390 | 5191 | 5734 | 6265 | 466 | 4426 | 5578 | 8322 | Statistics |
|---|---|---|---|---|---|---|---|---|---|---|---|
| Selected LAEs | \multicolumn: Identical (L18H4, L23H24, L21H21, L22H7) | | | | | | | | | | **100% Match** |
| AUC (%) | 87.05 | 87.12 | 87.18 | 87.14 | 87.09 | 87.21 | 87.16 | 87.10 | 87.13 | 87.19 | **$87.15 \pm 0.04$** |

ing variant, which applies a single, learnable channel-wise scaling vector to LAE features, improves performance by +2.88% AUC. Employing a more expressive additive steering mechanism, which instead modulates features with a learnable additive bias vector, further boosts the AUC to 85.02%. However, these limited gains underscore the inherent constraints of context-unaware modulation. Relying solely on the local gating module (LGM) to perform its context-dependent anisotropic scaling, without the global scrutiny gate (GSG), results in a 1.21% AUC degradation from our full model. This highlights the GSG's critical function as a high-level gate that determines if modulation is necessary, thereby preventing the LGM from altering features of benign frames indiscriminately. This result demonstrates that the synergistic interplay between the global gating mechanism and the local, fine-grained modulator is crucial for optimal performance.

**Effectiveness of Representational Separability Analysis.** We validate our RSA against several heuristic selection strategies, with results shown in Part B from Table 2. Choosing heads at random or first-to-mid layers yield poor performance due to their failure to capture task-relevant semantics. Positional heuristics selecting heads from mid-to-last layers provide better results. The late-layer heads perform better due to their encoding of higher-level semantic features, but the selection lacks precision. Our RSA method, which directly optimizes for geometric separability, significantly outperforms all baselines. This performance advantage stems from RSA's ability to measure the manifold disentanglement of each head rather than relying on qualitative architectural assumptions. By filtering for subspaces where normal and anomalous patterns naturally diverge, RSA ensures that the subsequent steering mechanism operates on the most geometrically compliant representations. The selected experts represent stable, task-specific functional circuits intrinsic to the frozen backbone, distinguishing them from transient or noisy features often selected by heuristic methods.

**Calibration Stability and Robustness.** To empirically verify that these identified circuits rely on intrinsic model properties rather than artifacts of the limited 1% calibration data, we conducted a rigorous stability analysis. As detailed in Table 3, we performed 10 independent calibration runs using different random seeds. The results demonstrate exceptional robustness: the RSA algorithm identified the strictly identical set of Latent Anomaly Experts (specifically heads L18H4, L23H24, L21H21, and L22H7) across all 10 runs, resulting in negligible AUC fluctuation ($\sigma \approx 0.04\%$). Furthermore, comparative analysis reveals that increasing data coverage from 1% to 100% yields marginal gains, confirming that the geometric signatures of these experts are sufficiently distinct to be robustly captured even with minimal supervision. This definitive consistency proves that our expert selection mechanism is invariant to specific data splits, establishing a solid and reproducible foundation for the subsequent geometric rectification.

**Table 4:** Data efficiency analysis. The rapid performance saturation indicates that the geometric steering policy converges to an optimal solution with minimal supervision (1%).

| Data Ratio | AUC (%) | Improvement | Training Time |
|:---:|:---:|:---:|:---:|
| **1% (Ours)** | **87.15** | - | **< 1 min** |
| 10% | 87.29 | +0.14 | ∼ 5 min |
| 25% | 87.35 | +0.20 | ∼ 13 min |
| 50% | 87.39 | +0.24 | ∼ 25 min |
| 100% | 87.42 | +0.27 | ∼ 49 min |

**Table 5:** Hyperparameter analysis of selected LAEs ($K$) and input frames ($F$) on UCF-Crime.

| Number of Experts ($K$) | | Number of Frames ($F$) | |
|:---:|:---:|:---:|:---:|
| **Value** | **AUC (%)** | **Value** | **AUC (%)** |
| 2 | 85.91 | 4 | 82.54 |
| **4** | **87.15** | 8 | 85.03 |
| 8 | 87.09 | **16** | **87.15** |
| 16 | 86.88 | 24 | 87.23 |

**Data Efficiency and Geometric Saturation.** To rigorously quantify the calibration efficiency, we systematically investigate the performance trajectory as the calibration set size scales from the minimal 1% subset ($N \approx 16$ videos) up to the full 100% training set. As detailed in Table 4, the model exhibits a distinct phenomenon of asymptotic saturation. While increasing the data coverage from 1% to 100% causes the computational cost to escalate linearly from less than one minute to approximately 49 minutes, the resulting performance gain is statistically negligible, yielding only a +0.27% improvement in AUC. This sharp plateau indicates that the HMC does not rely on massive data density to memorize patterns; instead, it rapidly converges to an optimal steering policy by capturing the intrinsic functional circuits of the backbone. This confirms that the geometric signature of anomalies is a low-rank property that can be robustly estimated with minimal supervision. Visual evidence in Appendix further corroborates this stability, showing that the spatial distribution of identified experts remains invariant across all data scales.

**Hyperparameter Analysis** We evaluate two critical hyperparameters in SteerVAD: the number of selected latent anomaly experts ($K$) and sampled frames ($F$) from each segment. As detailed in Table 5, the performance exhibits a clear parabolic relationship with $K$; the AUC peaks at 87.15% when $K = 4$ by striking an optimal balance between incorporating diverse, anomaly-centric features and mitigating the noise introduced by less discriminative experts at higher values. The choice of $F$ reveals a classic pattern of diminishing returns. Extending the input frames to 16 yields a substantial performance gain by integrating richer contextual information, this improvement saturates thereafter offering negligible further gains that do not justify the considerable increase in computational over­head. Therefore, we adopt $K = 4$ and $F = 16$ as our final configuration, representing the most compelling trade-off between model efficacy and computational efficiency.

## 5 CONCLUSION

In this paper, we introduce a novel intervention framework for tuning-free video anomaly detection, moving from passive feature interpretation to active geometric rectification within frozen MLLMs. Our SteerVAD operationalizes this by using representational separability analysis (RSA) to pin­point task-aligned latent anomaly experts and employing a hierarchical meta-controller (HMC) to dynamically rectify their representation manifolds through context-aware anisotropic scaling. This active steering mechanism effectively overcomes pre-training biases and resolves contextual ambi­guities, leading to new state-of-the-art performance on the UCF-Crime and XD-Violence bench­marks among tuning-free methods. Our work demonstrates that targeted, dynamic intervention is a powerful and data-efficient alternative to costly fine-tuning, paving the way for more adaptable applications of frozen foundation models for video anomaly detection.

## ACKNOWLEDGEMENTS

This work was supported in part by the National Natural Science Foundation of China under Grant 62225112, 62595732, 62401365, 62271312, 62132006, U24A20220, in part by the Sichuan Science and Technology Program under Grant 2026YFHZ0205, and in part by the XJTU Research Fund for Al Science, and in part by the China Postdoctoral Science Foundation under Grants BX20250411, 2025M773473.

## ETHICS STATEMENT

Our work strictly adheres to the ICLR Code of Ethics. The research is conducted exclusively on publicly available, standard academic benchmarks (UCF-Crime and XD-Violence), which are widely used for evaluating video anomaly detection algorithms. We acknowledge the dual-use nature of video anomaly detection technology. However, our primary motivation is to advance the field for societal benefit, such as enhancing public safety through intelligent surveillance and improving industrial quality control. We believe that our framework represents a step towards more responsible AI by promoting transparency and intervention accountability. Unlike opaque fine-tuning methods that alter millions of parameters, our approach isolates intervention to a small, identifiable set of latent anomaly experts (LAEs). This architectural choice makes the model's decision-making process more scrutable. Furthermore, our post-hoc explanation capability is not merely a descriptive feature; it provides a user-facing check on the system's outputs, allowing human operators to verify alerts against generated textual rationales. By focusing on rectifying pre-training biases at a geometric level, our work also contributes to mitigating one of the core ethical challenges in deploying large models. We advocate for the responsible development and deployment of this technology, ensuring its application aligns with ethical principles and contributes positively to human well-being.

## REPRODUCIBILITY STATEMENT

We are fully committed to ensuring the complete reproducibility of our research. To this end, we provide an exhaustive description of our methodology in Section 3, supplemented by a rigorous theoretical analysis in Appendix A and detailed implementation specifics, including architectural layouts and tensor transformations, in Appendix B. All crucial implementation details, including the specific frozen MLLM backbone used (InternVL3), dataset preprocessing steps, and a complete list of hyperparameters, are clearly documented in Section 4 and Appendix C. Our experiments are built on standard public benchmarks and common libraries (PyTorch, Hugging Face Transformers) to ensure broad accessibility. Furthermore, to significantly enhance the verifiability of our results, we will provide comprehensive JSON file samples of our experimental run outputs in the supplementary materials. We also pledge to release our source code upon the publication of this paper, which we believe will allow the community to fully replicate our findings and extend our work.

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

# A    THEORETICAL ANALYSIS OF THE REPRESENTATION MANIFOLD

This section formalizes the theoretical underpinnings of our geometric steering methodology. We argue that the emergence of structured, low-dimensional representational geometries is a natural consequence of applying continuous deep networks to high-dimensional, semantically coherent data. Our argument proceeds by first establishing the topological properties of the input data space, then analyzing how these properties are preserved under the MLLM's feature mapping.

## A.1    TOPOLOGICAL PROPERTIES OF THE INPUT DATA SPACE

We begin by characterizing the topological structure of the data space $\mathcal{V} \subset \mathbb{R}^N$ from which video segments are drawn. A semantic class $\mathcal{C}$ (e.g., "normal" or "anomaly") corresponds to a subset of this space, denoted $\mathcal{V}_{\mathcal{C}} \subset \mathcal{V}$.

**Compactness.**    We model the data subset $\mathcal{V}_{\mathcal{C}}$ for any given class $\mathcal{C}$ as a compact set. This model is grounded in two properties. First, video data is inherently bounded; pixel intensities are confined to a finite range (e.g., $[0, 255]$) and video dimensions are fixed, making $\mathcal{V}_{\mathcal{C}}$ a bounded subset of $\mathbb{R}^N$. Second, while the semantic boundary between classes can be ambiguous in practice, we posit that the underlying generative process for a class occupies a topologically closed region in the data space. This idealized assumption is a necessary abstraction for analytical tractability, allowing us to model the core data distribution as a well-defined set. Under the Heine-Borel theorem, a subset of $\mathbb{R}^N$ that is both closed and bounded is compact. This compactness provides a stable topological foundation, ensuring that continuous maps such as the attention heads in an MLLM exhibit well-behaved properties when applied to the data.

**Piecewise Path-Connectedness.**    The data structure for complex semantic categories is inherently piecewise, not monolithic. A high-level concept such as "anomaly" encompasses a diverse family of distinct scenarios; for instance, "arson" in a building and a "vehicle collision" on a street are both anomalies, yet no continuous transformation path of semantically valid videos exists in the high-dimensional data space to connect them.

This intrinsic heterogeneity dictates that the data space for a class $\mathcal{C}$ is most accurately modeled as a union of path-connected subsets. Formally, we express this structure as:

$$\mathcal{V}_{\mathcal{C}} = \bigcup_{i=1}^{M_{\mathcal{C}}} \mathcal{V}_{\mathcal{C}}^{(i)} \tag{10}$$

where each component $\mathcal{V}_{\mathcal{C}}^{(i)}$ is a path-connected and compact set, representing a semantically coherent subcluster (e.g., "theft" and "vandalism" as components of "anomaly"). Crucially, these components are not necessarily disjoint; a single event, such as breaking a car window to steal an item, can naturally belong to multiple subclusters. This potential for overlap in the input space is a critical feature, as it implies that the resulting representation manifolds in the latent space may also intersect, creating a complex, entangled geometry that is difficult to separate.

This topological structure has a direct implication for the learned representation space. As the feature extractor $h$ is a continuous map, it preserves path-connectedness. Consequently, the image of $\mathcal{V}_{\mathcal{C}}$ under $h$, which we term the representation manifold $\mathcal{M}_{\mathcal{C}} = h(\mathcal{V}_{\mathcal{C}})$, is also a collection of path-connected manifold components:

$$\mathcal{M}_{\mathcal{C}} = \bigcup_{i=1}^{M_{\mathcal{C}}} h\left(\mathcal{V}_{\mathcal{C}}^{(i)}\right) = \bigcup_{i=1}^{M_{\mathcal{C}}} \mathcal{M}_{\mathcal{C}}^{(i)} \tag{11}$$

Here we formally define the representation manifold $\mathcal{M}_{\mathcal{C}}$ as this complete, set-theoretic union. Crucially, the MLLM's pre-training on vast semantic data induces a critical property in this structure: the manifold components $\{\mathcal{M}_{\mathcal{C}}^{(i)}\}$ are not scattered arbitrarily across the latent space but are mapped into a bounded and largely contiguous region. This geometric coherence ensures that the global properties of the parent manifold $\mathcal{M}_{\mathcal{C}}$ specifically its centroid (center of mass) are not just well-defined, but serve as a geometrically meaningful representation of the class's central tendency. Consequently, our

RSA metric is not a naive simplification but a robust measure of the separation between the centers of these high-level structures.

This leads to a more precise geometric formulation of the VAD challenge: the task is not to separate two single, cleanly-defined manifolds, but to distinguish between two families of potentially overlapping and entangled manifold components $\{\mathcal{M}_{\text{norm}}^{(i)}\}$ and $\{\mathcal{M}_{\text{anom}}^{(j)}\}$.

## A.2 FEATURE EXTRACTORS AS CONTINUOUS MAPS

An MLLM feature extractor, such as an attention head, is modeled as a function $h : \mathcal{V} \to \mathbb{R}^{d_{\text{head}}}$. This function is a composition of a finite number of layers $h = f_L \circ f_{L-1} \circ \cdots \circ f_1$, each consisting of linear transformations and continuous, non-linear activation functions (e.g., GELU, Softmax). Since the composition of continuous functions is itself continuous, the entire feature extractor $h$ constitutes a continuous map from the input space $\mathcal{V}$ to the representation space $\mathbb{R}^{d_{\text{head}}}$.

A fundamental theorem in topology states that the continuous image of a path-connected set is path-connected. Given our model of the input data space where each component $\mathcal{V}_\mathcal{C}^{(i)}$ is path-connected and the map $h$ is continuous, it follows that the image of each component, $\mathcal{M}_\mathcal{C}^{(i)} = h(\mathcal{V}_\mathcal{C}^{(i)})$, must also be a path-connected subset of $\mathbb{R}^{d_{\text{head}}}$. This formalizes the observation that feature representations of semantically coherent subclusters coalesce into distinct geometric structures within the representation space, collectively forming the full representation manifold $\mathcal{M}_\mathcal{C}$.

## A.3 LOCAL EUCLIDEAN STRUCTURE AND INTRINSIC DIMENSIONALITY

For $\mathcal{M}$ to qualify as a differentiable manifold, it must also be locally homeomorphic to a Euclidean space. This property emerges from the smooth mapping of local degrees of freedom in the input data to the representation space.

Under the central tenet of the manifold hypothesis, we assume that local variations within a small neighborhood of a video $v \in \mathcal{C}$ (e.g., an object's position, orientation) can be parameterized by a low-dimensional vector $\mathbf{u} \in U \subset \mathbb{R}^{d_\mathcal{M}}$, where $U$ is an open set and $d_\mathcal{M} \ll N$ is the number of local generative factors. This defines video in this neighborhood as a local parameterization $v(\mathbf{u})$.

The feature extractor $h$ acts on this parameterization, defining a local coordinate chart $\phi : U \to \mathbb{R}^{d_{\text{head}}}$ via the composition:

$$\phi(\mathbf{u}) = h(v(\mathbf{u})) \tag{12}$$

Since $h$ is composed of differentiable (or piecewise differentiable) functions, the map $\phi$ is also differentiable (or piecewise differentiable). This map embeds a patch from the low-dimensional parameter space $\mathbb{R}^{d_\mathcal{M}}$ as a surface within the high-dimensional representation space $\mathbb{R}^{d_{\text{head}}}$.

The local geometry of this embedded surface is characterized by the Jacobian of the chart:

$$J_\phi(\mathbf{u}) = \frac{\partial \phi}{\partial \mathbf{u}} \in \mathbb{R}^{d_{\text{head}} \times d_\mathcal{M}} \tag{13}$$

The $d_\mathcal{M}$ column vectors of the Jacobian span the tangent space $T_p\mathcal{M}$ at a point $p = \phi(\mathbf{u})$. We operate under the reasonable assumption that for non-degenerate natural variations, this map is an immersion, meaning the Jacobian has full column rank, i.e., $\text{rank}(J_\phi) = d_\mathcal{M}$. This non-collapsing condition ensures that the local dimensionality is preserved. The existence of such a differentiable atlas (a collection of charts covering $\mathcal{M}$) establishes $\mathcal{M}$ as a differentiable manifold.

## A.4 IMPLICATIONS AND RATIONALE FOR GEOMETRIC STEERING

**Geometric Problem Formulation.** The preceding analysis provides a rigorous geometric framework for understanding the VAD task and justifying our intervention strategy. From this perspective, the VAD challenge arises when the representation manifolds for normal ($\mathcal{M}_{\text{norm}}$) and anomalous ($\mathcal{M}_{\text{anom}}$) data are proximal or intersect in $\mathbb{R}^{d_{\text{head}}}$. Such proximity, where $\inf_{\mathbf{p} \in \mathcal{M}_{\text{norm}}, \mathbf{q} \in \mathcal{M}_{\text{anom}}} \|\mathbf{p} - \mathbf{q}\|_2 \approx 0$, is often due to shared local features (e.g., "a person running" is common to both jogging and fleeing). This entanglement makes reliable separation by a static classifier difficult.

**Intervention as a Controlled Duality of Transformation.** As introduced in the methodology, our hierarchical meta-controller (HMC) learns a context-dependent transformation $T_{\mathbf{c}} : \mathbb{R}^{d_{\text{head}}} \rightarrow \mathbb{R}^{d_{\text{head}}}$ via anisotropic manifold scaling. This transformation exhibits a powerful duality:

- **Diffeomorphic Remodeling.** When all scaling factors are non-zero, $T_{\mathbf{c}}$ acts as a local diffeomorphism, smoothly remodeling the geometric structure by anisotropically stretching and compressing the manifold along discriminative axes while preserving local topology. This alters the local metric tensor, effectively increasing the geometric distance between manifolds without tearing their topology.

- **Singular Projection.** The true expressive power of our method is unleashed when the framework drives one or more scaling factors to zero. This induces a singularity, rendering the transformation locally non-invertible and fundamentally changing its nature from remodeling to structural simplification. Geometrically, this singular transformation acts as a projection, mapping the feature space $\mathbb{R}^{d_{\text{head}}}$ onto a lower-dimensional effective subspace spanned by the dimensions with non-zero scaling. Information within the projection's null space, the dimensions corresponding to zero scaling, is actively discarded. This mechanism serves as a form of potent, context-aware feature selection, allowing the model to completely nullify dimensions associated with pre-training biases or task-irrelevant variance. This projection effectively creates what can be conceptualized as a stable anchor space; representations are anchored to this simpler subspace, and their final structure becomes robust to perturbations in the suppressed dimensions, whose outputs are mapped to a stable fixed point (e.g., the origin) within the projected subspace.

This capability justifies our choice of anisotropic manifold scaling. It is not merely a heuristic but a highly expressive yet efficient transformation capable of learning both smooth, geometry-preserving diffeomorphisms and radical, context-aware projections for dimensional reduction. This provides a minimal yet powerful mechanism to achieve the desired manifold separation.

**Theoretical Objective and Practical Realization.** The ultimate objective of our adaptive transformation is to maximize the geometric separation between transformed manifolds, thereby resolving structural ambiguity in their latent spaces. This objective can be formally expressed as maximizing a metric like the Hausdorff distance, $d_H(T_{\mathbf{c}}(\mathcal{M}_{\text{norm}}), T_{\mathbf{c}}(\mathcal{M}_{\text{anom}}))$. While direct optimization of such a metric is often intractable, our practical training objective, binary cross-entropy (BCE) loss serves as an effective and stable surrogate. By penalizing misclassifications, the BCE loss implicitly compels the HMC to learn a transformation $T_{\mathbf{c}}$ that renders the manifolds maximally separable to a linear classifier. This aligns the tractable optimization process with our fundamental geometric goal, training an expressive transformation to achieve a well-defined structural objective.

This concludes the theoretical derivation, which demonstrates that representation manifolds are a predictable consequence of applying continuous deep networks to structured data and provides a rigorous foundation for our geometric steering approach.

## A.5 TOPOLOGICAL GENERALIZATION TO OPEN-SET SCENARIOS

A fundamental consideration in applying the manifold hypothesis to real-world surveillance is the unbounded, open-set nature of video data. While the input space of raw pixels $\mathcal{V}$ is theoretically infinite, we posit that the latent representation space induced by a pre-trained MLLM imposes a structured topology that facilitates generalization to unseen events. Here, we extend our theoretical framework to formally account for open-set robustness.

**Open-Set as Modular Component Expansion.** Recall from Eq. 11 that the anomaly manifold $\mathcal{M}_{anom}$ is modeled not as a monolithic shape, but as a union of semantically coherent path-connected components: $\mathcal{M}_{anom} = \bigcup_i \mathcal{M}_{anom}^{(i)}$. Under open-set conditions, an unseen anomaly type does not violate this manifold structure; rather, it manifests as the emergence of a novel topological component, $\mathcal{M}_{new}^{(k)}$. Critically, due to the generalized discriminative capacity of the foundation model, this new component $\mathcal{M}_{new}^{(k)}$ is mapped to a region topologically distinct from the normality manifold $\mathcal{M}_{norm}$. Our framework relies on the principle of geometric invariance, while the intrinsic coordinates of $\mathcal{M}_{new}^{(k)}$ are unknown during calibration, its geometric divergence from the normality baseline remains a consistent, detectable property.

**RSA as Geometric Consistency Filter.** The Representational Separability Analysis (RSA) operationalizes this principle by functioning as a geometric filter. Rather than searching for attention heads that classify specific known anomalies, RSA identifies functional subspaces where the topological separation between the support of $\mathcal{M}_{norm}$ and any divergent signal is naturally maximized. Subspaces exhibiting chaotic mapping where open-set samples might collapse into the normality manifold are assigned low separability scores and are strictly excluded from the intervention scope.

**Entropy-Driven Steering Policy.** Consequently, the intervention policy learned by the hierarchical meta-controller (HMC) is not one of closed-set classification, but of context-relative divergence correction. Conditioned on the global context $c$, the HMC identifies directions in the tangent space $T_p\mathcal{M}$ that exhibit high deviation from the expected context coherence (analogous to high semantic entropy). This formulation explains the model's ability to handle unseen data, novel anomalies act as high-entropy perturbations that trigger the pre-learned steering mechanism to project them away from $\mathcal{M}_{norm}$, regardless of their specific semantic content.

## B    DETAILS OF STEERVAD

This section provides a detailed description of the implementation of our proposed SteerVAD framework, complementing the methodology presented in Section 3. We elaborate on the precise mathematical formulations, the architectural specifications of our trainable modules, and the algorithmic procedures for latent anomaly expert (LAE) discovery, model training, and real-time inference.

### B.1    PREPARATION OF STEERVAD

#### B.1.1    REPRESENTATIONAL SEPARABILITY ANALYSIS (RSA)

The representational separability analysis (RSA) is a systematic, gradient-free procedure designed to identify a select set of latent anomaly experts (LAEs). The objective is to pinpoint attention heads whose internal feature spaces are most naturally aligned with the VAD task, meaning they maximize a criterion of class separability. This section provides the detailed mathematical formulation and algorithmic implementation, complementing the conceptual overview in Section 3. The complete procedure is formally outlined in Algorithm 1.

At its core, RSA leverages a geometric separability criterion inspired by the Fisher Discriminant Ratio to evaluate and rank the representational efficacy of attention heads. This metric is deliberately chosen for its computational tractability, as it operates solely on first and second order statistics (mean and variance), and circumvents the need to train auxiliary classifiers per head, which would incur prohibitive computational and optimization overhead. More fundamentally, its mathematical objective of maximizing between-class scatter while minimizing within-class scatter directly aligns with our architectural goal: to steer the manifold and rectify them for unbiased detection. An attention head that scores highly under this criterion induces a feature geometry in which intra-class compactness and inter-class margin are simultaneously optimized, rendering the representation ideally suited for subsequent classification by a simple and parameter-efficient linear scorer.

For each attention head $h_{l,k}$ (at layer $l$, index $k$), its RSA score is computed as the ratio of the between-class scatter to the total within-class scatter:

$$S_{\text{RSA}}(l,k) = \frac{S_B(l,k)}{S_W(l,k) + \epsilon} \tag{14}$$

where $\epsilon$ is a small constant for numerical stability.

The numerator, $S_B(l,k)$, represents the between-class scatter. It is defined as the squared Euclidean distance between the centroids of the anomalous and normal feature distributions, rewarding heads that map the two classes to distant locations in the embedding space. It is calculated as:

$$S_B(l,k) = \|\boldsymbol{\mu}_{\text{anom}}^{(l,k)} - \boldsymbol{\mu}_{\text{norm}}^{(l,k)}\|_2^2 \tag{15}$$

The denominator, $S_W(l,k)$, represents the total within-class scatter. It quantifies the compactness of the clusters by summing the squared Euclidean distances of each feature vector to its respective class centroid. A lower value indicates more tightly formed, less diffuse clusters. It is defined as:

$$S_W(l, k) = \sum_{\mathbf{x} \in \mathcal{D}_{\text{anom}}^{(l,k)}} \|\mathbf{x} - \boldsymbol{\mu}_{\text{anom}}^{(l,k)}\|_2^2 + \sum_{\mathbf{x} \in \mathcal{D}_{\text{norm}}^{(l,k)}} \|\mathbf{x} - \boldsymbol{\mu}_{\text{norm}}^{(l,k)}\|_2^2 \quad (16)$$

where $\mathcal{D}_{\text{norm/anom}}^{(l,k)}$ denotes the set of feature vectors $\{\mathbf{x}\}$ produced by head $(l, k)$ for all normal and anomalous samples in the calibration set, where each $\mathbf{x} \in \mathbb{R}^{d_{\text{head}}}$. The term $\boldsymbol{\mu}$ represents the centroid (mean vector) of the corresponding feature set.

---

**Algorithm 1** Representational Separability Analysis (RSA) for LAE Discovery

---

1: **Input:** Frozen MLLM $\mathcal{M}$, calibration dataset $\mathcal{D}_{\text{calib}}$.
2: **Parameter:** Number of experts to select $K$.
3: **Output:** A set of $K$ Latent Anomaly Experts (LAEs).

4:                                       ▷ *Phase 1: Prepare a class-balanced calibration set*
5: Construct the calibration set $\mathcal{D}_{\text{calib}}$ to balanced by randomly undersampling the majority class.

6:                                       ▷ *Phase 2: Feature Extraction*
7: Initialize empty feature banks for all heads $(l, k)$:
8: $\mathcal{B}_{\text{norm}}^{(l,k)} \leftarrow \emptyset$, $\mathcal{B}_{\text{anom}}^{(l,k)} \leftarrow \emptyset$
9: **for** each video segment $S_t$ with label $y_t$ in $\mathcal{D}_{\text{calib}}$ **do**
10:      Perform a single forward pass of $S_t$ through $\mathcal{M}$.
11:      For each attention head $(l, k)$, extract its output feature vector $\mathbf{h}_{l,k}^{(t)}$.
12:      **if** $y_t$ is normal **then**
13:          Add $\mathbf{h}_{l,k}^{(t)}$ to $\mathcal{B}_{\text{norm}}^{(l,k)}$.
14:      **else**
15:          Add $\mathbf{h}_{l,k}^{(t)}$ to $\mathcal{B}_{\text{anom}}^{(l,k)}$.
16:      **end if**
17: **end for**

18:                                          ▷ *Phase 3: Score Calculation*
19: Initialize an empty list of scores $\mathcal{S} \leftarrow []$.
20: **for** each attention head $(l, k)$ **do**
21:      Compute centroids: $\boldsymbol{\mu}_{\text{norm}}^{(l,k)} = \text{mean}(\mathcal{B}_{\text{norm}}^{(l,k)})$, $\boldsymbol{\mu}_{\text{anom}}^{(l,k)} = \text{mean}(\mathcal{B}_{\text{anom}}^{(l,k)})$.
22:      Compute between-class scatter $S_B(l, k) = \|\boldsymbol{\mu}_{\text{anom}}^{(l,k)} - \boldsymbol{\mu}_{\text{norm}}^{(l,k)}\|_2^2$.
23:      Compute within-class scatters:
24:      $S_{W,\text{norm}} = \sum_{\mathbf{h} \in \mathcal{B}_{\text{norm}}^{(l,k)}} \|\mathbf{h} - \boldsymbol{\mu}_{\text{norm}}^{(l,k)}\|_2^2$.
25:      $S_{W,\text{anom}} = \sum_{\mathbf{h} \in \mathcal{B}_{\text{anom}}^{(l,k)}} \|\mathbf{h} - \boldsymbol{\mu}_{\text{anom}}^{(l,k)}\|_2^2$.
26:      Compute RSA score $S_{\text{RSA}}(l, k) = S_B(l, k)/(S_{W,\text{norm}} + S_{W,\text{anom}} + \epsilon)$.
27:      Add $(l, k, S_{\text{RSA}}(l, k))$ to $\mathcal{S}$.
28: **end for**

29:                                            ▷ *Phase 4: LAE Selection*
30: Sort $\mathcal{S}$ in descending order based on $S_{\text{RSA}}$.
31: **return** the top $K$ heads from the sorted list $\mathcal{S}$.

---

**Equivalence to the Variance-Based Metric.** To ensure robust and unbiased estimation, we compute the RSA score on a class-balanced calibration set, constructed by randomly undersampling the majority class to match the size of the minority class. This procedural detail is critical as it establishes a direct mathematical link between the scatter-based formula used for implementation (Eq. 14) and the more intuitive variance-based metric presented in Methods (Eq. 1).

Specifically, the variance of a cluster is defined as $\sigma^2 = \frac{1}{N} \sum_{\mathbf{x}} \|\mathbf{x} - \boldsymbol{\mu}\|_2^2$. Consequently, the within-class scatter for a class is $S_W^{\text{class}} = N \cdot \sigma^2$. On a balanced dataset, the total within-class scatter becomes:

$$S_W(l, k) = N \cdot \sigma_{\text{anom}}^2(l, k) + N \cdot \sigma_{\text{norm}}^2(l, k) = N \left( \sigma_{\text{anom}}^2(l, k) + \sigma_{\text{norm}}^2(l, k) \right) \quad (17)$$

Substituting this into our RSA score gives:

$$S_{\text{RSA}}(l, k) = \frac{\|\boldsymbol{\mu}_{\text{anom}}^{(l,k)} - \boldsymbol{\mu}_{\text{norm}}^{(l,k)}\|_2^2}{N \left(\sigma_{\text{anom}}^2(l, k) + \sigma_{\text{norm}}^2(l, k)\right) + \epsilon} \tag{18}$$

Since our goal is to rank the attention heads, maximizing this score is equivalent to maximizing the metric from Eq. 1, as $N$ is a positive constant that does not affect the ranking order. This balancing act prevents the statistics of a majority class from dominating the score and justifies our use of the more computationally direct scatter formulation.

This principled, quantitative analysis allows us to survey all attention heads within the frozen MLLM and rigorously select the top-$K$ performers. These selected heads, designated as latent anomaly experts, are not just randomly chosen sub-modules; they are empirically validated as the most informative and geometrically aligned subspaces for the VAD task, making them the optimal targets for our subsequent manifold rectification.

### B.1.2 HIERARCHICAL META-CONTROLLER (HMC) AND ANOMALY SCORER

This section provides detailed implementations of trainable components in SteerVAD. We move beyond the conceptual overview in Section 3 to detail the precise tensor transformations, architectural choices, and design rationales essential for reproducibility. The dimensions specified here correspond to our experiments using the InternVL model, where the hidden dimension is $d_{\text{model}} = 3584$ and the attention head dimension is $d_{\text{head}} = 128$.

**Global Context Vector Rationale**    The efficacy of our context-aware rectification hinges upon the extraction of the global context vector, $\mathbf{c} \in \mathbb{R}^{d_{\text{model}}}$. As introduced in Section 3.4, this vector is derived from the final hidden state of the frozen MLLM at the first generated token. This choice is motivated by the auto-regressive nature of the underlying language model. The model generates the initial token by constructing a holistic semantic summary through comprehensive encoding of the entire input sequence, yielding a hidden state that encapsulates global contextual understanding. Further from a structural standpoint, the causal attention mechanism ensures that this representation is purely conditioned on the inputs, free from the influence of any generated text. This yields an unbiased and information-rich summary of the scene's semantics, providing an ideal input for the HMC while remaining computationally efficient as a natural byproduct of the single forward pass.

**Global Scrutiny Gate (GSG)**    The global scrutiny gate is used to distill the high-dimensional global context vector $\mathbf{c}$ into a single suspicion score $s_{\text{global}}$. It processes a batch of context vectors of shape $[B, d_{\text{model}}]$ to produce a batch of scores of shape $[B, 1]$

**Table 6:** Architecture of the global scrutiny gate (GSG). The input batch size is denoted by $B$.

| Layer | Input Shape | Output Shape | Activation / Notes |
|-------|-------------|--------------|--------------------|
| Input $\mathbf{c}$ | $[B, 3584]$ | - | Global context vector |
| Linear-1 | $[B, 3584]$ | $[B, 128]$ | - |
| BatchNorm1d | $[B, 128]$ | $[B, 128]$ | Stabilizes training |
| ReLU | $[B, 128]$ | $[B, 128]$ | Non-linearity |
| Linear-2 | $[B, 128]$ | $[B, 1]$ | Output logit |
| Sigmoid | $[B, 1]$ | $[B, 1]$ | Produces $s_{\text{global}}$ |

**Architecture.**    The module is implemented as a lightweight MLP, whose architecture is detailed in Table 6. The inclusion of a `BatchNorm1d` layer is a deliberate choice to stabilize the training dynamics on the small calibration set. It normalizes the feature activations from the first linear layer, preventing the model from becoming reliant on specific dimensions of the context vector whose statistics might be inconsistent across small batches.

**Local Gating Module (LGM)**    The local gating module is designed for parameter-efficient, context-dependent feature modulation. It consists of $K$ parallel low-rank adapters, where $K$ is the number of selected LAEs. Each adapter takes the same global context vector $\mathbf{c}$ and generates a unique steering vector $\mathbf{g}_i$ for its corresponding LAE.

**Table 7:** Architecture of a single low-rank adapter within the LGM.

| Layer | Input Shape | Output Shape | Activation / Notes |
|---|---|---|---|
| Input $\mathbf{c}$ | $[B, 3584]$ | - | Shared across $K$ adapters |
| Linear-down ($r = 4$) | $[B, 3584]$ | $[B, 4]$ | `bias=False` |
| Linear-up | $[B, 4]$ | $[B, 128]$ | `bias=False` |
| Tanh | $[B, 128]$ | $[B, 128]$ | Produces $\mathbf{g}_i \in [-1, 1]$ |

**Architecture.** Each of the $K$ adapters is identical in structure, as detailed in Table 7. The design choice to set `bias=False` in both linear layers is intentional. It ensures that the adapter's operation is purely modulatory (scaling existing features) rather than additive (shifting features), preserving the original manifold's position while only altering its geometry. This aligns with our core objective of rectifying, not replacing the learned representations of MLLM. Numerical computations are performed using `bfloat16` precision, consistent with the native format of the InternVL backbone.

---

**Algorithm 2** Training Procedure for HMC and Anomaly Scorer

---

1: **Input:** A batch of pre-extracted LAE features, context vectors, and labels $\{(\{\mathbf{h}_i\}_t, \mathbf{c}_t, y_t)\}_{t=1}^B$.
2: **Parameters:** HMC parameters $\theta_{\text{HMC}}$, Scorer parameters $\theta_{\text{Scorer}}$, regularization weight $\lambda_{\text{reg}}$.
3: Initialize batch loss $\mathcal{L}_{\text{batch}} \leftarrow 0$.
4: **for** each sample $(\{\mathbf{h}_i\}, \mathbf{c}, y)$ in the batch **do**
5:                                                         ▷ *Generate Rectification Signals*
6:     Compute global suspicion score $s_{\text{global}} = \text{GSG}(\mathbf{c}; \theta_{\text{HMC}})$.
7:     Compute local steering vectors $\{\mathbf{g}_i\}_{i=1}^K = \text{LGM}(\mathbf{c}; \theta_{\text{HMC}})$.
8:                                          ▷ *Perform Anisotropic Manifold Scaling*
9:     Compute rectified features: $\mathbf{h}_i' = \mathbf{h}_i \odot (1 + s_{\text{global}} \cdot \mathbf{g}_i)$ for $i = 1, \ldots, K$.
10:                                       ▷ *Score and Compute Loss*
11:     Concatenate features: $F_{\text{final}} = \text{Concat}(\mathbf{h}_1', \ldots, \mathbf{h}_K')$.
12:     Compute logit $z = \text{AnomalyScorer}(F_{\text{final}}; \theta_{\text{Scorer}})$.
13:     Compute classification loss $\mathcal{L}_{\text{BCE}} = \text{BCEWithLogitsLoss}(z, y)$.
14:                         ▷ *Compute Regularization Loss for Normal Samples*
15:     Initialize $\mathcal{L}_{\text{reg}} \leftarrow 0$.
16:     **if** $y$ is normal **then**
17:         $\mathcal{L}_{\text{reg}} = (s_{\text{global}})^2$.
18:     **end if**
19:     Accumulate total loss: $\mathcal{L}_{\text{batch}} \leftarrow \mathcal{L}_{\text{batch}} + \mathcal{L}_{\text{BCE}} + \lambda_{\text{reg}} \mathcal{L}_{\text{reg}}$.
20: **end for**
21: Perform backpropagation on $\mathcal{L}_{\text{batch}}$ to compute gradients $\nabla_{\theta_{\text{HMC}}, \theta_{\text{Scorer}}} \mathcal{L}_{\text{batch}}$.
22: Update parameters $\theta_{\text{HMC}}$ and $\theta_{\text{Scorer}}$ using an optimizer (e.g., Adam).

---

**Anisotropic Manifold Scaling** The core rectification mechanism, anisotropic scaling, is an element-wise tensor operation that integrates the outputs of the GSG and LGM to modulate the raw LAE features. The data flow for a batch of inputs during inference is as follows:

1. **Inputs**: The process begins with two tensors: the raw LAE features $\mathbf{H} \in \mathbb{R}^{B \times K \times d_{\text{head}}}$ and the global context vectors $\mathbf{c} \in \mathbb{R}^{B \times d_{\text{model}}}$.

2. **Signal Generation**: The context $\mathbf{c}$ is passed through the GSG to produce the global scores $\mathbf{s}_{\text{global}} \in \mathbb{R}^{B \times 1}$, and concurrently through the $K$ adapters of the LGM to produce the stacked local steering vectors $\mathbf{G} \in \mathbb{R}^{B \times K \times d_{\text{head}}}$.

3. **Rectification**: The global score $\mathbf{s}_{\text{global}}$ is unsqueezed to a shape of $[B, 1, 1]$ to enable broadcasting across the $K$ experts and $d_{\text{head}}$ dimensions. The final rectified features $\mathbf{H}'$ are computed via the element-wise operation:

$$\mathbf{H}' = \mathbf{H} \odot (1 + \text{unsqueeze}(\mathbf{s}_{\text{global}}) \cdot \mathbf{G}) \tag{19}$$

where $\odot$ denotes the Hadamard product. This mechanism provides dynamic, fine-grained control over the feature geometry. For instance, a high global score ($s_{\text{global}} \approx 1$) combined with positive steering signals in $\mathbf{G}$ amplifies the norm of corresponding feature vectors, while negative signals attenuate them, thus actively steering the representation on its manifold.

---

**Algorithm 3** Real-time Inference Pipeline of SteerVAD

---

1: **Input:** Untrimmed video stream $\mathcal{V}$; Pre-trained HMC and Anomaly Scorer; LAE configuration.
2: **Parameters:** Frames per segment $L$; Sampled frames per segment $F$; Gaussian kernel std. dev. $\sigma_g$; Anomaly threshold $\tau_{\text{anomaly}}$.
3: **Output:** Smoothed anomaly curve $\mathbf{A}_{\text{smooth}}$; Set of textual explanations $\mathcal{E}$ (optional).
4:                                              ▷ *Phase 1: Frame-level Probability Scoring*
5: Partition $\mathcal{V}$ into $T$ non-overlapping segments $\{S_1, \ldots, S_T\}$, each of length $L$.
6: Initialize an empty list of probabilities $\mathcal{P} \leftarrow []$.
7: **for** each segment $S_t$ from $t = 1$ to $T$ **do**
8:      Uniformly sample $F$ frames from $S_t$.
9:      Perform a single forward pass through the frozen MLLM to extract raw LAE features $\{\mathbf{h}_i\}_t$ and context $\mathbf{c}_t$.
10:      With gradients disabled:
11:          $s_{\text{global}}^{(t)} \leftarrow \text{GSG}(\mathbf{c}_t)$,     $\{\mathbf{g}_i\}_t \leftarrow \text{LGM}(\mathbf{c}_t)$.
12:          $\mathbf{h}_i' = \mathbf{h}_i \odot (1 + s_{\text{global}}^{(t)} \cdot \mathbf{g}_i)$ for $i = 1, \ldots, K$.
13:          $F_{\text{final}}^{(t)} = \text{Concat}(\mathbf{h}_1', \ldots, \mathbf{h}_K')$.
14:          $z_t = \text{AnomalyScorer}(F_{\text{final}}^{(t)})$.
15:      $p_t = \sigma(z_t)$, where $\sigma$ is the Sigmoid function.
16:      Append $p_t$ to $\mathcal{P}$.
17: **end for**
18:                                          ▷ *Phase 2: Anomaly Curve Generation*
19: Let $N_{frames}$ be the total number of frames in $\mathcal{V}$. Initialize frame-level scores $\mathbf{A}_{\text{raw}} \in \mathbb{R}^{N_{frames}}$ with zeros.
20: **for** $t = 1$ to $T$ **do**
21:      Let $[start, end)$ be the frame indices for segment $S_t$.
22:      $\mathbf{A}_{\text{raw}}[start : end] = p_t$.            ▷ Assign segment probability to corresponding frames
23: **end for**
24: Compute smoothed curve $\mathbf{A}_{\text{smooth}} = \text{GaussianConvolution1D}(\mathbf{A}_{\text{raw}}, \sigma_g)$.

25:                                      ▷ *Phase 3: Post-Hoc Explanation (Optional)*
26: Initialize an empty set of explanations $\mathcal{E} \leftarrow \emptyset$.
27: **for** $t = 1$ to $T$ **do**
28:      Compute representative score for segment $S_t$, e.g., $\bar{a}_t = \text{mean}(\mathbf{A}_{\text{smooth}}[start : end])$.
29:      **if** $\bar{a}_t > \tau_{\text{anomaly}}$ **then**
30:          Let $V_t$ be the original frames of segment $S_t$.
31:          Construct prompt: $Q_t$.
32:          Generate explanation: $E_t = \text{MLLM.generate}(Q_t, V_t)$.
33:          Add $(t, E_t)$ to $\mathcal{E}$.
34:      **end if**
35: **end for**
36: **return** $\mathbf{A}_{\text{smooth}}, \mathcal{E}$.

---

**Anomaly Scorer** The design of anomaly scorer prioritizes robustness over complexity. Its primary role is to map the rectified feature space to a final anomaly probability.

**Architecture and Rationale.** The scorer first reshapes the batch of rectified features $\mathbf{H}' \in \mathbb{R}^{B \times K \times d_{\text{head}}}$ into a flattened feature matrix $\mathbf{F}_{\text{final}} \in \mathbb{R}^{B \times (K \cdot d_{\text{head}})}$. For our configuration with $K = 4$, this results in an input shape of $[B, 512]$. This matrix is then processed by a single linear layer (`nn.Linear(512, 1)`) to produce a batch of logits $\mathbf{z} \in \mathbb{R}^{B \times 1}$.

This minimalist design, equivalent to a logistic regression classifier, is a deliberate methodological choice that acts as a form of regularization. By constraining the classifier's capacity, we ensure that performance gains are attributable to the quality of the rectified representations produced by the HMC, rather than the power of the final classifier. It compels the HMC to learn a feature space that is as linearly separable as possible.

**Training Procedure** The HMC and anomaly scorer are trained end-to-end using the composite loss function from Equation 9. The complete training process is detailed in Algorithm 2.

### B.2 Real-time Inference of SteerVAD

The inference pipeline of SteerVAD is designed for efficiency, enabling the processing of long, untrimmed videos in a single pass and culminating in a final, interpretable output. The entire process, formally detailed in Algorithm 3, integrates segment-wise scoring, temporal aggregation for curve generation, and an optional mechanism for generating textual explanations for detected anomalies.

The procedure begins by partitioning the input video into a sequence of non-overlapping temporal segments. Each segment is processed independently through a single forward pass of the frozen MLLM to extract raw LAE features and a global context vector. The pre-trained HMC and Scorer modules then compute a segment-level anomaly probability. This raw sequence of probabilities, while informative, can exhibit high-frequency noise due to transient visual fluctuations.

To address this and produce a temporally coherent result, we introduce an aggregation step. The segment-level scores are first mapped to a frame-level representation. Subsequently, we apply a temporal smoothing filter, as implemented in our visualization code via a standard 1D Gaussian convolution. This operation, which efficiently realizes the weighted averaging formula in Equation 6, smooths the raw scores to produce the final anomaly curve, effectively suppressing spurious spikes while preserving sustained anomalous events.

Finally, to enhance the framework's utility and trustworthiness, an optional post-hoc explanation module can be activated. For any segment where the smoothed anomaly score surpasses a predefined threshold, the corresponding video frames are re-submitted to the frozen MLLM's generative interface. By prompting the MLLM to describe the events in these frames, we obtain a natural language explanation for the detected anomaly, bridging the gap between a numerical score and a human-understandable reason.

## C Further Experiment Details

This section provides additional details regarding our experimental setup to ensure full reproducibility. We outline the specific implementation environment, the prompts used to query the MLLM, and a comprehensive summary of all hyperparameters used in our framework.

### C.1 Implementation Details

**Model and Architecture.** Our experiments are conducted using the InternVL3-8B (Zhu et al., 2025) model as the frozen MLLM backbone. This model is based on the InternViT-300M-448px-V2.5 vision encoder and a Qwen2.5-7B-based language model (Qwen et al., 2025). The language model comprises 28 transformer layers, with each layer containing 28 attention heads. This results in a total of $N_{\text{total}} = 28 \times 28 = 784$ attention heads available for the representational separability analysis (RSA). The hidden dimension of each attention head is $d_{\text{head}} = 128$.

**Hardware and Software.** All experiments were performed on a single server running Ubuntu 22.04.4 LTS. The hardware configuration features dual AMD EPYC 75F3 CPUs (128 logical cores), 755 GB of system RAM, and a single NVIDIA RTX A6000 GPU with 48 GB of VRAM. Our implementation is built upon PyTorch (v2.3.0) and the Hugging Face Transformers library (v4.51.3), utilizing CUDA 12.1 for GPU acceleration.

### C.2 Prompt Details

During the development of our framework, we explored several prompts to elicit the most effective global context vector from the MLLM. Below are the main variants we considered. For all experiments reported in this paper, we utilized the structured system prompt (Prompt 3), which we found to yield the most consistent and task-aligned representations for anomaly detection.

**Prompt 1 (General Query).**

"If there are any abnormal events that occurred in the video that deviate from the normal scene, please identify them and describe what happened."

**Prompt 2 (Structured Response Query).**

"Please carefully watch the video and identify if any abnormal events occur. Your task is to first provide a clear and concise summary of the video content, then state whether any mentioned abnormal events are present. If yes, specify which events occurred and provide a detailed description of what happened, including the timing and context. If no, explain why you believe the video is normal. Please structure your response as follows:

- Yes or No answer first with anomaly detection.
- **Video Summary:** A brief summary of the video content.
- **Details:** If Yes, describe the abnormal events in detail. If No, explain your reasoning.

Take your time to analyze the video thoroughly and ensure your assessment is accurate. Take a deep breath and work on this problem step-by-step."

**Prompt 3 (Final System Prompt).**

**System Identity**
You are a professional AI assistant specialized in video analysis, focusing on abnormal behavior detection in real-time surveillance.

**Task Context**
Analyse if there are abnormal events happened in video frames.

**Core Objectives**

1. Analyze the following video frames.
2. Detect high-risk abnormal events which should not happen in this video.
3. Perform spatiotemporal analysis across consecutive frames.

**Output Requirements**
Implement a two-level response system with Yes or No first, then subsequent paragraphs must include: Detected event type(s), Detailed description of the behavior, and Risk level assessment (low/medium/high).

## C.3 HYPERPARAMETER CONFIGURATION

Table 8 provides a comprehensive summary of all hyperparameters used throughout our framework, including those for LAE discovery, HMC training, and real-time inference.

## D FURTHER EXPERIMENTS

### D.1 PERFORMANCE EVALUATION

To provide a comprehensive understanding of the practicality of our framework, this section details the computational performance of SteerVAD. We evaluate the model size, training resource requirements, and the inference costs associated with the trainable components, demonstrating that our active intervention approach is not only effective but also remarkably efficient.

**Model Size and Complexity.** The primary advantage of our tuning-free framework is the minimal size of the trainable modules. The hierarchical meta-controller (HMC) and the anomaly scorer operate on top of a frozen, multi-billion parameter MLLM, yet their own footprint is negligible. Table 9 summarizes the architectural and size statistics of these components, based on the final configuration used in our experiments ($K = 4$ experts). The total number of trainable parameters is approximately 0.52 million, which occupies about 1 MB of disk space. This is several orders

**Table 8:** Consolidated list of hyperparameters for the SteerVAD framework.

| Hyperparameter | Value | Description |
|---|---|---|
| ***Data Processing & Model Architecture*** | | |
| $L$ | 48 frames | The duration of each non-overlapping video segment. |
| $F$ | 16 frames | The number of frames sampled from each segment as input to the MLLM. |
| $K$ | 4 | The number of top-scoring latent anomaly experts (LAEs) selected by RSA. |
| ***HMC Training*** | | |
| Epochs | 1000 | The total number of training epochs for the HMC and anomaly scorer. |
| Batch Size | 64 | The number of samples per batch during training. |
| Learning Rate | $1 \times 10^{-3}$ | The initial learning rate for the Adam optimizer. |
| $\lambda_{\mathrm{reg}}$ | 0.1 | The weight for the sparsity regularization loss ($\mathcal{L}_{\mathrm{reg}}$). |
| $d_{\mathrm{hidden}}$ | 128 | The hidden dimension of the MLP in the Global Scrutiny Gate (GSG). |
| $r$ | 4 | The low-rank dimension of the adapters in the Local Gating Module (LGM). |
| ***Inference & Analysis*** | | |
| $\sigma_g$ | 6.0 | The standard deviation of the 1D Gaussian kernel for temporal smoothing. |
| $\tau_{\mathrm{anomaly}}$ | 0.5 | The score threshold to trigger the optional post-hoc explanation module. |
| $\epsilon$ | $1 \times 10^{-8}$ | A small constant for numerical stability in the RSA score calculation. |

of magnitude smaller than the frozen backbone, highlighting the parameter-efficient nature of our method. The computational overhead is similarly minimal, requiring just over 1 million FLOPs per segment inference, ensuring that the bottleneck remains the single forward pass through the MLLM.

**Table 9:** Architectural size and computational complexity of SteerVAD's trainable modules.

| Component | Parameters | Size / Complexity |
|---|---|---|
| ***Parameter Count & Disk Size*** | | |
| Global Scrutiny Gate | 459,265 | — |
| Local Gating Module | 59,392 | — |
| **HMC (Total)** | **518,657** | **0.99 MB** |
| Anomaly Scorer | 513 | <0.01 MB |
| **Total Trainable** | **519,170** | **˜0.99 MB** |
| ***Computational Complexity (per inference)*** | | |
| HMC FLOPs | | 1,037,568 |
| Scorer FLOPs | | 1,024 |
| **Total Added FLOPs** | | **1,038,592** |

**Table 10:** Training cost statistics for calibrating the HMC and Anomaly Scorer.

| Metric | Value |
|---|---|
| Total Training Epochs | 1000 |
| Batch Size | 64 |
| Learning Rate | 0.001 |
| **Total Training Time** | **27.83 seconds (0.46 minutes)** |
| Average Time per Epoch | 0.03 seconds |
| Peak GPU Memory Usage | 203.88 MB |
| Training Throughput | 4168.81 samples/second |

**Training Cost.** The calibration of the HMC and anomaly scorer is extremely efficient. The training is performed on a small, representative calibration set randomly sampled 1% videos from the training set from UCF-Crime and Xd-Violence. As detailed in Table 10, the entire training process completes in approximately one minute on a single commodity GPU. The memory footprint is minimal, and the training throughput is exceptionally high. This low-cost calibration phase makes SteerVAD easy to adapt to new domains or datasets without incurring the prohibitive expenses associated with fine-tuning large foundation models.

**Table 11:** Inference cost as a function of the number of sampled frames ($F$) per segment.

| Sampled Frames ($F$) | Avg. Inference Time (s) | Peak VRAM (GB) |
|:---:|:---:|:---:|
| 4 | 0.43 | 17.2 |
| 8 | 0.78 | 18.3 |
| 16 | 1.52 | 20.4 |
| 24 | 2.29 | 22.3 |

**Inference Cost Analysis.** The primary factor influencing inference cost in our framework is the number of frames, $F$, sampled per video segment to be processed by the MLLM. Table 11 presents the trade-off between the temporal density of this sampling and the resulting computational demands in terms of latency and GPU memory. As expected, both inference time and peak VRAM usage scale near-linearly with the number of sampled frames. This analysis demonstrates a clear and predictable trade-off, allowing practitioners to balance detection performance with their available computational budget. The configuration used in our main experiments ($F = 16$) was chosen to provide a robust temporal context while remaining well within the capacity of common research hardware.

**Table 12:** Cross-dataset generalization comparison. To ensure fairness, we reproduce HiProbeVAD under identical settings with InternVL3-8B backbone with same prompt1 setting in App.C.2.

| Source (Calibration) | Target (Evaluation) | Metric | HiProbeVAD (Rep) | SteerVAD (Ours) | Gap |
|---|---|---|---|---|---|
| UCF-Crime (1%) | XD-Violence (Unseen) | AP (%) | 70.66 | **71.31** | +0.65 |
| XD-Violence (1%) | UCF-Crime (Unseen) | AUC (%) | 80.82 | **81.04** | +0.22 |

**Zero-Shot Generalization.** We further evaluate the robustness of SteerVAD by testing its zero-shot transfer capability across distinct domains. In this setting, the HMC module is calibrated on a source dataset and directly applied to a completely unseen target dataset. To provide a rigorous benchmark, we compare our method against the state-of-the-art probing baseline, HiProbe-VAD, implemented with the same backbone and settings. As presented in Table 12, SteerVAD demonstrates consistent superiority over the baseline in both transfer directions, retaining high performance despite significant shifts in anomaly definitions and environmental contexts. This cross-dataset stability indicates that the hierarchical meta-controller does not overfit to superficial domain-specific cues within the small calibration set. Instead, it effectively learns a transferable geometric rectification policy, identifying and amplifying high-entropy divergences that remains valid across diverse video distributions.

**Table 13:** Cross-model generalization performance. The framework is applied to different most widely used and advanced frozen MLLM backbones.

| Backbone | Params | UCF-Crime (AUC %) | XD-Violence (AP %) |
|---|---|---|---|
| LLaVA-OV (Li et al., 2024) | ~7B | 81.52 | 74.65 |
| Qwen2.5-VL (Bai et al., 2025) | ~7B | 84.11 | 79.82 |
| **InternVL3 (Zhu et al., 2025)** | **~8B** | **87.15** | **83.02** |

**Cross-Model Generalization.** Our active manifold steering strategy is a fundamental principle rather than a technique coupled to a specific architecture. We applied the SteerVAD framework to other prominent MLLMs, as presented in Table 13, different models including LLaVA-OV and Qwen2.5VL with our steering mechanism unlocks strong anomaly detection capabilities. On both the UCF-Crime and XD-Violence datasets, these models achieve a level of performance that is highly competitive. This outcome powerfully suggests that the existence of latent, geometrically steerable expert subspaces is a general property of MLLMs. SteerVAD framework should be understood as a model-agnostic paradigm for adapting a wide range of frozen foundation models to complex perception tasks, rather than a solution tied to a single model.

# E EXTENDED ROBUSTNESS AND GENERALIZATION ANALYSIS

In this section, we present a comprehensive empirical evaluation of the SteerVAD framework's adaptability. Building on the data efficiency analysis in the main text, we focus on the model's sensitivity to class distribution imbalances, the structural stability of the expert selection process under random initialization, and its capability to generalize to strictly unseen open-set scenarios. Furthermore, we substantiate these quantitative results with extensive visual evidence of the underlying latent geometric structures.

## E.1 VISUALIZATION OF GEOMETRIC SATURATION

While the quantitative saturation is discussed in the main paper, we provide further intuitive visualization of the RSA score distribution across all attention heads ($L \times H$). To demonstrate the universality of our findings, we generate these heatmaps for all 14 categories within the UCF-Crime dataset under varying calibration set sizes. As illustrated in Figure 5 and 6, the heatmaps of anomaly sensitivity exhibit remarkable invariance. The high-scoring regions, which represent the latent anomaly experts, emerge distinctly at the 1% scale and maintain their spatial positions and relative intensities as data scales to 100%. This visual evidence, consistent across diverse anomaly semantics, powerfully corroborates that the geometric signature of anomalies is a stable, low-rank property, confirming that massive calibration data is redundant for identifying the core functional circuits.

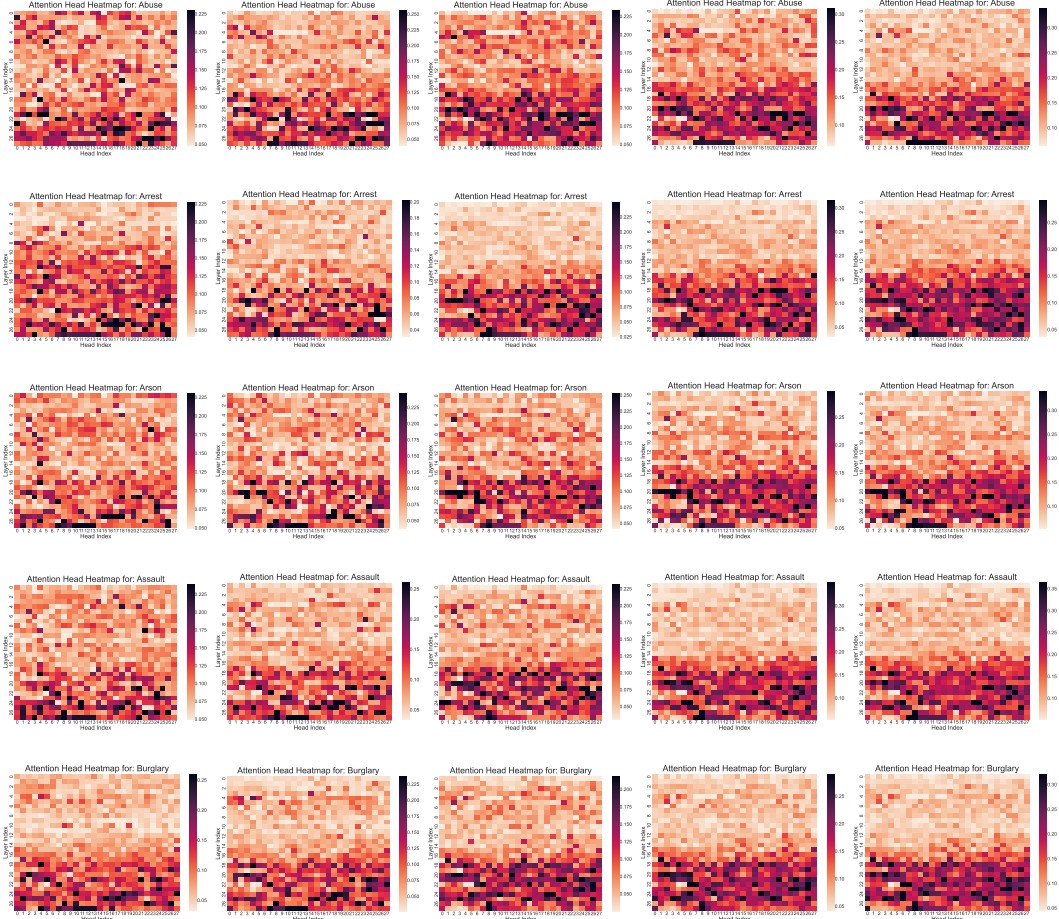

**Figure 5:** Evolution of anomaly expert activations across different calibration data scales. The consistent distribution of hotspots from 1% to 100% visually confirms that the identification of anomaly experts saturates rapidly.

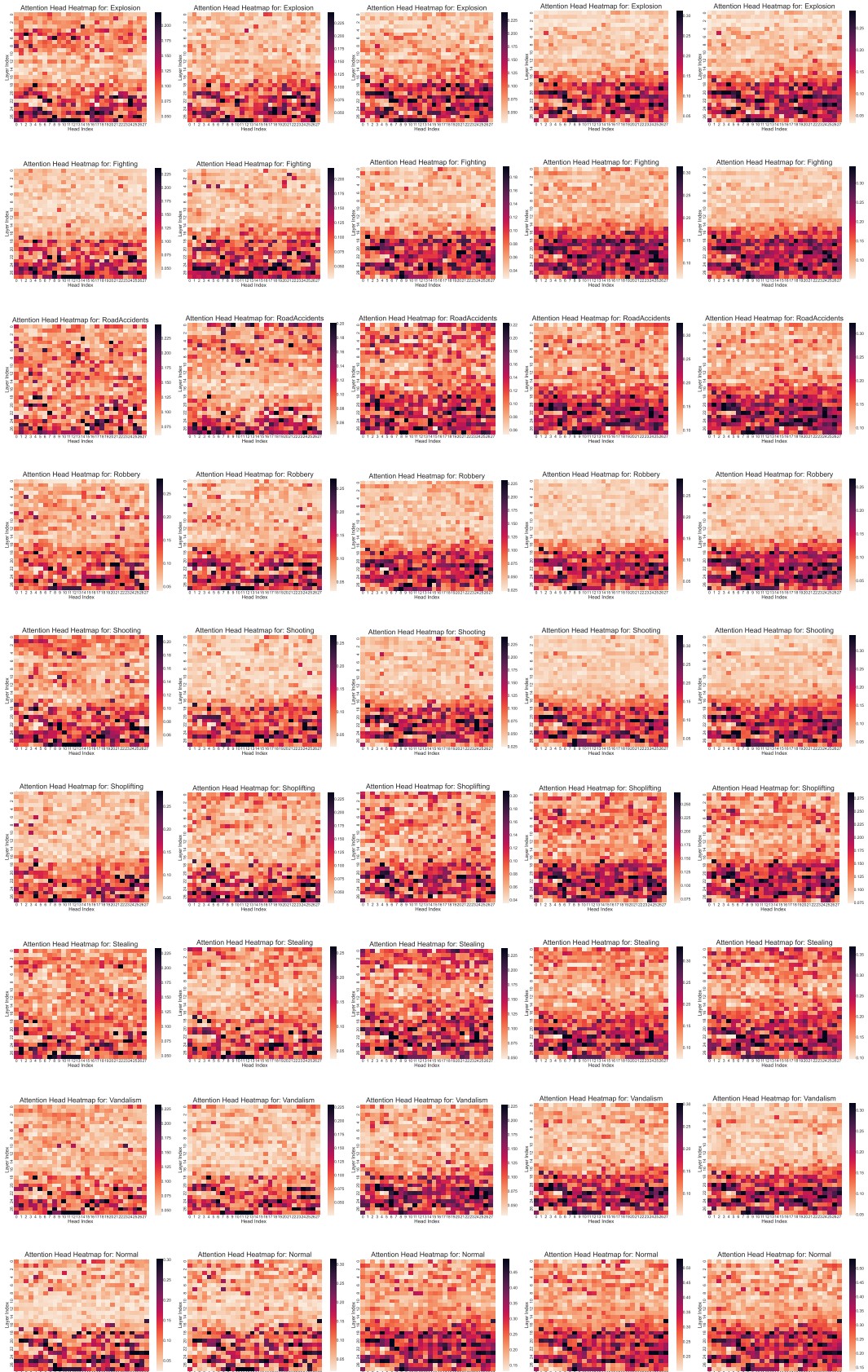

**Figure 6:** Evolution of anomaly expert activations across different calibration data scales. The consistent distribution of hotspots from 1% to 100% visually confirms that the identification of anomaly experts saturates rapidly.

**Table 14:** Impact of normal/abnormal ratio in the calibration set on UCF-Crime performance. The model maintains robust detection capabilities across a wide range of class distributions.

| Ratio (N:A) | 1:9 | 2:8 | 3:7 | 4:6 | 5:5 (Default) | 6:4 | 7:3 | 8:2 | 9:1 |
|---|---|---|---|---|---|---|---|---|---|
| **AUC (%)** | 83.67 | 85.69 | 85.75 | 86.34 | **87.15** | 87.00 | 85.70 | 85.62 | 83.72 |

### E.2 SENSITIVITY TO CLASS DISTRIBUTION IMBALANCE

The robustness of the hierarchical meta-controller (HMC) against label distribution shifts is a critical factor for real-world deployment. We systematically analyzed the sensitivity of our framework to varying ratios of normal versus abnormal videos in the calibration set. Theoretically, to learn a discriminative steering vector $g_i$ that effectively projects anomalous features away from the normality manifold, the HMC requires a contrastive set of examples to model the direction of geometric divergence. As detailed in Table 14, empirical results demonstrate significant resilience. While a balanced distribution (5:5) yields the peak performance, the method maintains robust AUC scores above 86% across a wide effective range (from 4:6 to 6:4). Performance degradation becomes notable only at extreme imbalances. In such regimes, the scarcity of counter-examples prevents the HMC from establishing a stable normality baseline, leading to a less precise rectification policy.

Crucially, this finding highlights a distinct advantage of our few-shot calibration paradigm in handling the inherent sparsity of real-world anomalies. Unlike fully supervised methods that require massive, balanced datasets for convergence a condition rarely met in practice due to the long-tail nature of abnormal events, our framework requires only a minute, representative subset for calibration. Consequently, even in scenarios where anomalies are extremely rare in the wild, the ability to construct a small, locally balanced calibration set from limited samples allows SteerVAD to mitigate the data sparsity problem effectively. This capability provides a robust foundation for learning in extremely sparse scenarios, where traditional approaches would fail to acquire sufficient signal.

### E.3 ROBUSTNESS TO SUPERVISION GRANULARITY

To evaluate the robustness of SteerVAD under coarser supervision signals, we conducted ablation study using video-level labels. In this setting, all segments belonging to an anomalous video are labeled as positive during the calibration phase. As shown in Table 15, the model achieves 87.01% AUC with coarse video-level labels, which is statistically comparable to the 87.15% AUC obtained with standard segment-level labels. This minimal performance drop (-0.14%) demonstrates that the RSA-based expert selection and HMC modulation are highly robust to label noise, effectively distilling the anomaly signature even from coarse annotations.

**Table 15:** Impact of supervision granularity on UCF-Crime performance. SteerVAD maintains high performance even with coarse video-level labels.

| Supervision Type | Granularity | AUC (%) |
|---|---|---|
| **Segment-Level (Default)** | **Fine** | **87.15** |
| Video-Level | Coarse | 87.01 |

### E.4 STRUCTURAL STABILITY OF EXPERT SELECTION

While the main text confirmed the stability of final AUC performance across random seeds, we further scrutinized the structural consistency of the selected feature subspaces. Figure 7 presents the jaccard similarity matrix of the Top-20 ranked attention heads across 10 independent random seeds. The heatmap reveals a high degree of overlap between any pair of random seeds. This indicates that the set of identified functional circuits is not sensitive to specific data samples. The algorithm consistently converges to the same subspace of expert heads, reinforcing that RSA captures stable, intrinsic functional components of the frozen backbone rather than fitting to transient data noise.

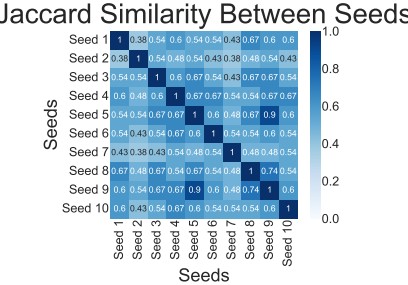

**Figure 7:** Jaccard similarity analysis of the Top-20 identified anomaly experts across 10 random seeds. The high similarity scores confirm that the expert selection is structurally robust to sampling variations.

**Table 16:** Detailed per-class AUC (%) performance on UCF-Crime. The active steering mechanism demonstrates consistent robustness across diverse anomaly types, with particularly high performance in dynamic events.

| Class | AUC (%) | Class | AUC (%) |
|---|---|---|---|
| Abuse | 68.84 | Robbery | 84.02 |
| Arrest | 82.58 | RoadAccident | 89.64 |
| Arson | 59.02 | Shooting | 85.74 |
| Assault | 95.17 | Shoplifting | 81.57 |
| Burglary | 75.47 | Stealing | 80.18 |
| Explosion | 79.72 | Vandalism | 88.16 |
| Fighting | 83.03 | **Average** | **87.15** |

### E.5 DETAILED ANALYSIS OF ANOMALY SEMANTICS AND FUNCTIONAL CIRCUITS

To investigate the operational boundaries and internal stability of our framework, we synthesize a granular performance analysis across all 13 anomaly classes (Table 16) with a visualization of the structural consistency of attention head selection (Figure 8).

**Geometric Divergence with Contextual Mimicry.** As detailed in Table 16, the framework exhibits performance heterogeneity that correlates with the semantic ambiguity of the event. Categories characterized by high-velocity motion and explicit visual conflict, such as Assault (95.17%), Road Accidents (89.64%), and Vandalism (88.16%), achieve peak AUC scores. These events generate representations with large displacement vectors relative to the normality manifold, which are easily amplified by our steering mechanism.

In contrast, categories such as Abuse (68.84%) and Burglary (75.47%) present a distinct challenge rooted in contextual mimicry. Visually, a burglary closely resembles a normal entry, and abuse often mimics normal physical interaction. These anomalies possess a smaller geometric footprint and high semantic overlap with normal activities. The difficulty here lies not merely in geometric separation, but in resolving the intent behind visually ambiguous actions, which suggests that future improvements for these classes may require incorporating longer-range temporal reasoning into the steering logic.

**Latent Structural Invariance.** To verify that the identified anomaly experts represent stable functional circuits rather than artifacts of random initialization, Figure 8 visualizes the selection frequency of top attention heads across 10 independent random seeds. The resulting heatmap reveals a striking structural invariance. A specific subset of heads is consistently selected with nearly identical across all runs, while the majority of the background heads remain inactive. This consistent convergence pattern demonstrates that RSA robustly identifies intrinsic functional anchors within the frozen MLLM that are uniquely sensitive to anomaly semantics, independent of the stochasticity in data sampling.

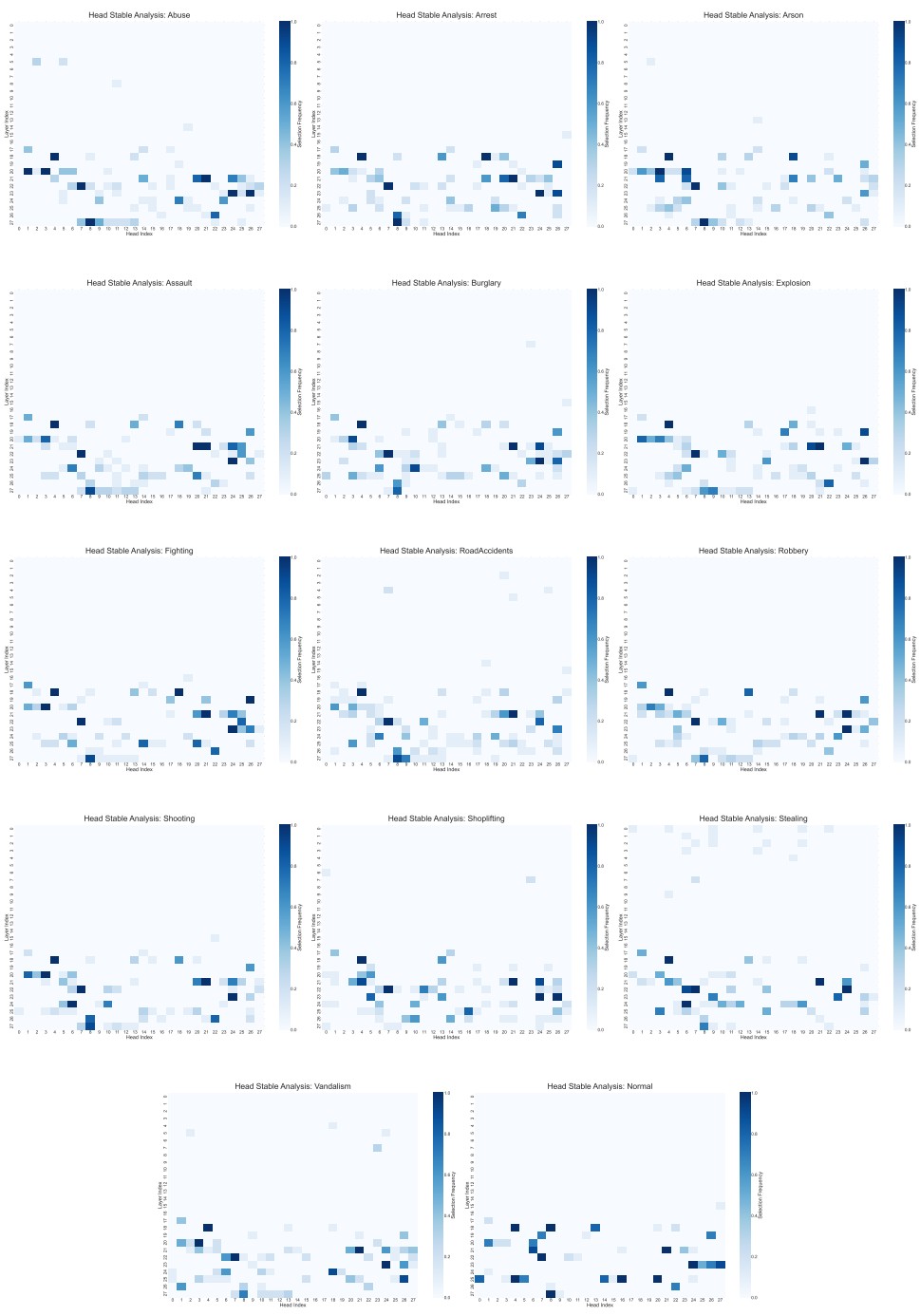

**Figure 8:** Visualization of top attention head selection frequency across 10 independent random seeds. The emergence of consistent high-frequency heads confirms that the model reliably identifies the same intrinsic functional circuits for anomaly detection, demonstrating robustness against random calibration splits.

### E.6 GENERALIZATION TO OPEN-SET AND UNSEEN ANOMALIES

To empirically validate the topological generalization framework proposed in Appendix A.5, we conducted a stress test on the UBnormal dataset. SteerVAD achieves a compelling **72.21% AUC** on this benchmark without any re-calibration. This result provides strong empirical evidence for our semantic entropy hypothesis, even for strictly unseen anomaly types, the anomalous events manifest as high-entropy geometric perturbations in the latent space, which are effectively rectified by our entropy-driven steering policy.

## F MECHANISM DESIGN VALIDATION

This section provides a deeper investigation into the architectural design choices of our framework, specifically validating the rationale behind using a linear metric for RSA and the architectural constraints of the HMC.

### F.1 VALIDITY OF LINEAR RSA VS. NON-LINEAR METRICS

To justify the use of a linear metric (RSA) for analyzing complex manifolds, we compared it against two computationally intensive metrics, Silhouette Coefficient and k-Nearest Neighbor (k-NN) Purity. We visualize the trade-off between computational cost and selection consistency in Figure 9.

The top panel highlights the dramatic efficiency gap that RSA completes calibration in milliseconds (0.14s), whereas k-NN incurs computational costs that are orders of magnitude higher (6.87s). Despite the speed difference, the head stability heatmaps for RSA, Silhouette, and k-NN are visually indistinguishable. All three metrics highlight the exact same expert regions. This visual evidence, combined with the strictly identical expert overlap shown in Table 17, conclusively proves that linearity is not a limitation but a feature, RSA serves as a lossless, high-speed proxy for identifying the manifold structure.

**Table 17:** Comparison of RSA with non-linear selection metrics regarding expert overlap and computational efficiency. RSA identifies the identical set of experts as expensive non-linear metrics but with significantly higher speed.

| Metric | Linearity | Expert Overlap | Calibration Time | Speedup |
|---|---|---|---|---|
| **RSA (Ours)** | **Linear** | **100% (Reference)** | **0.14 s** | **49×** |
| Silhouette | Non-Linear | 100% Identical | 0.86 s | 8× |
| k-NN Purity | Non-Linear | 100% Identical | 6.87 s | 1× |

### F.2 ABLATION OF HMC INPUT DESIGN

We explicitly designed the HMC to condition purely on the global context vector $c$. To verify this design, we implemented a local-aware variant where the input to the LGM adapter is the concatenation of the global context and the local feature ($[c, h_i]$).

As shown in Table 18, the local-aware variant results in a performance decrease of 0.80% AUC. This confirms that including the high-dimensional local feature causes the lightweight HMC to overfit to local noise. By restricting the input to the global context, we enforce a top-down modulation mechanism, ensuring that rectification signals are robust and context-driven.

## G QUALITATIVE ANALYSIS

**Visualization of Attention** The improved geometric organization of our SteerVAD corresponds to a refined semantic attention mechanism, as evidenced by the activation heatmaps in Figure 10. Initially, feature activations from the base model are scattered, indicating a failure to prioritize the semantically critical regions and an apparent distraction by biased objects within the video frames. Upon rectification by our framework, the focus of SteerVAD is unequivocally sharpened. The resulting feature activations are highly concentrated and precisely highlight the key anomalous entities,

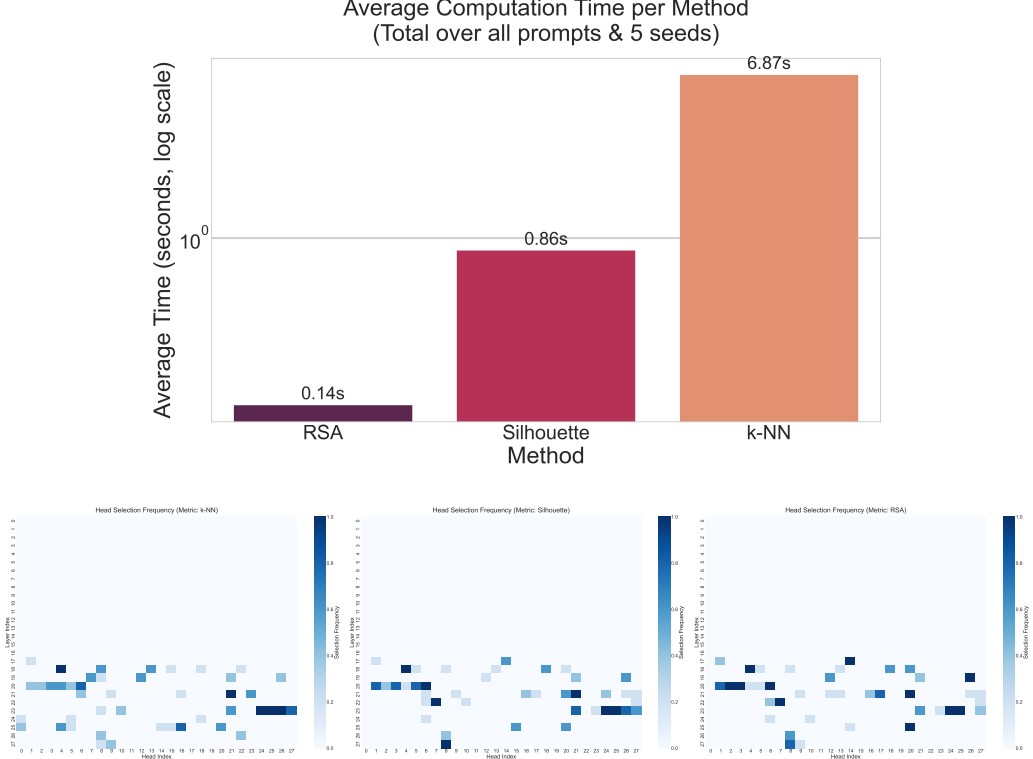

**Figure 9:** Validation of RSA efficiency and consistency. (Top) Average computation time per method, showing RSA's superior speed advantage. (Bottom) Heatmaps of head importance scores generated by RSA, Silhouette, and k-NN. The nearly identical visual patterns confirm that RSA captures the same underlying geometric structure as computationally expensive non-linear metrics.

**Table 18:** Design ablation for HMC input signals on UCF-Crime. Conditioning solely on global context prevents overfitting to local feature noise and ensures robust top-down modulation.

| Design Variant | Input Signal | AUC (%) | Observation |
|---|---|---|---|
| Local-Aware | $[c, h_i]$ (Concat) | 86.35 | Overfits to local noise & Self-Bias |
| **SteerVAD (Ours)** | **Global Context $c$ Only** | **87.15** | **Robust Top-Down Modulation** |

such as the central figures in the video. This demonstrates a core capability of our hierarchical meta-controller: it dynamically modulates the feature channels to amplify anomaly-relevant signals and attenuate noise, thereby correcting the semantic gaze of the model for improved task performance.

**Case Studies on Anomaly Detection.** Figure 11 demonstrates the end to end effectiveness and robustness of SteerVAD across normal and abnormal examples from both benchmark datasets. For an abnormal event in UCF-Crime, the anomaly score curve accurately rises at the moment of action, a conclusion supported by the generated explanation. Crucially, the model maintains a low score in a busy but normal surveillance scene, demonstrating resilience to false positives. This robustness extends to the challenging XD-Violence dataset. The model precisely captures abnormal event while correctly discerning a high motion sports clip as normal, a key challenge that confounds many methods. This ability to distinguish chaotic yet benign activity from genuine threats, coupled with the semantic coherence of the generated explanations, confirms that our framework delivers not only quantitatively superior results but also practically reliable and interpretable detections.

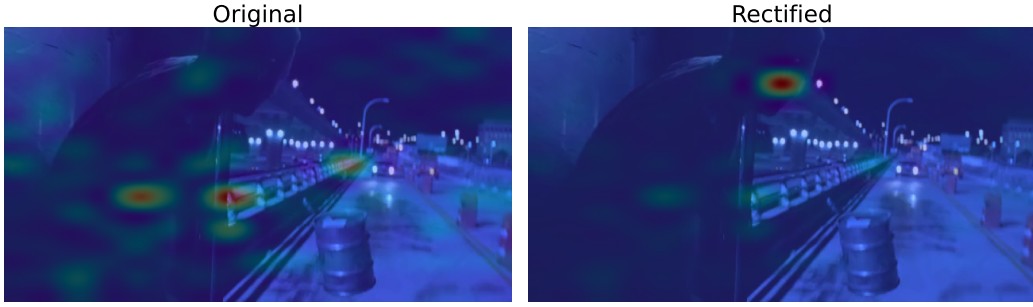

**Figure 10:** Semantic impact of feature rectification on an example from XD-Violence.

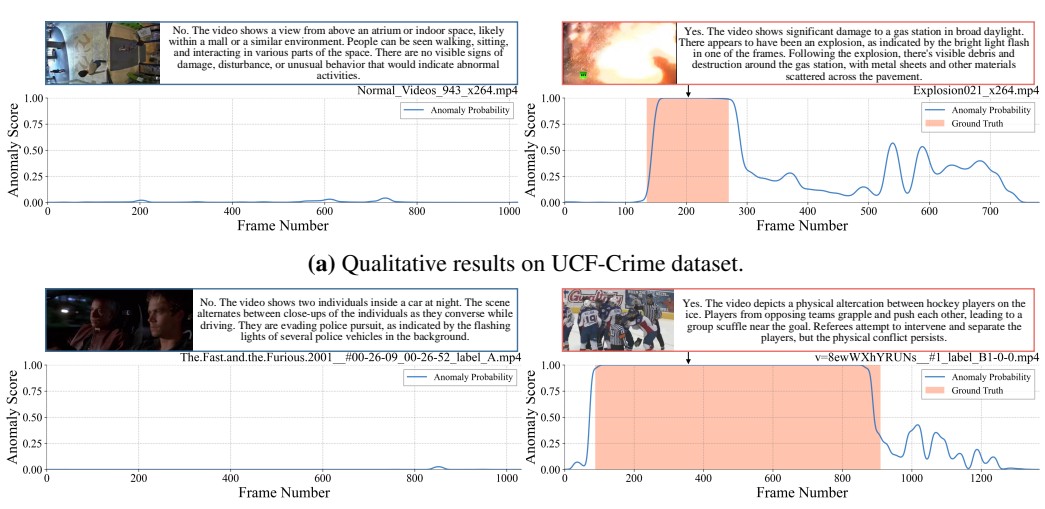

**Figure 11:** Qualitative results of SteerVAD on UCF-Crime and XD-Violence datasets. Each panel shows a representative video snippet and the anomaly curve with one normal and one abnormal video. The shaded regions denote the ground truth. Further generated descriptions are also provided.

## G.1 ADDITIONAL VISUALIZATIONS

To further illustrate the effectiveness and interpretability of our framework, we provide additional qualitative results and attention visualizations.

Figures 12 and 13 present a comprehensive selection of qualitative results on the UCF-Crime and XD-Violence datasets, respectively showcasing representative examples from all anomaly categories within each dataset. These examples reinforce the findings from our main case studies and demonstrate the consistent ability of SteerVAD to generate precise anomaly scores and coherent textual explanations across a wide variety of complex scenes. The inclusion of all categories highlights the model's robustness in correctly identifying diverse anomalies while avoiding false positives in dynamic but normal environments, further validating the practical reliability of our method.

Furthermore, Figures 14 and 15 offer more visualizations the semantic impact of our manifold rectification process. We visualize the model's effective gaze both before and after intervention. The original focus heatmaps frequently reveal the MLLM's inherent biases, showing attention scattered across irrelevant background elements or non-discriminative objects. In stark contrast, the rectified focus heatmaps demonstrate a clear and decisive shift. Our HMC successfully steers the model's attention to the core anomalous action, providing compelling visual evidence that our method not only improves geometric separability but also enforces a more task-relevant semantic understanding.

## H    LLM USAGE STATEMENT

In the preparation of this manuscript, the large language model (LLM) is solely utilized for the purpose of language polishing and grammar correction to improve clarity and readability. We confirm that all core research ideas, the conceptual framework, experimental design, data analysis are entirely the original work of the authors. The authors take full and sole responsibility for the accuracy, integrity, and all claims made within this work.

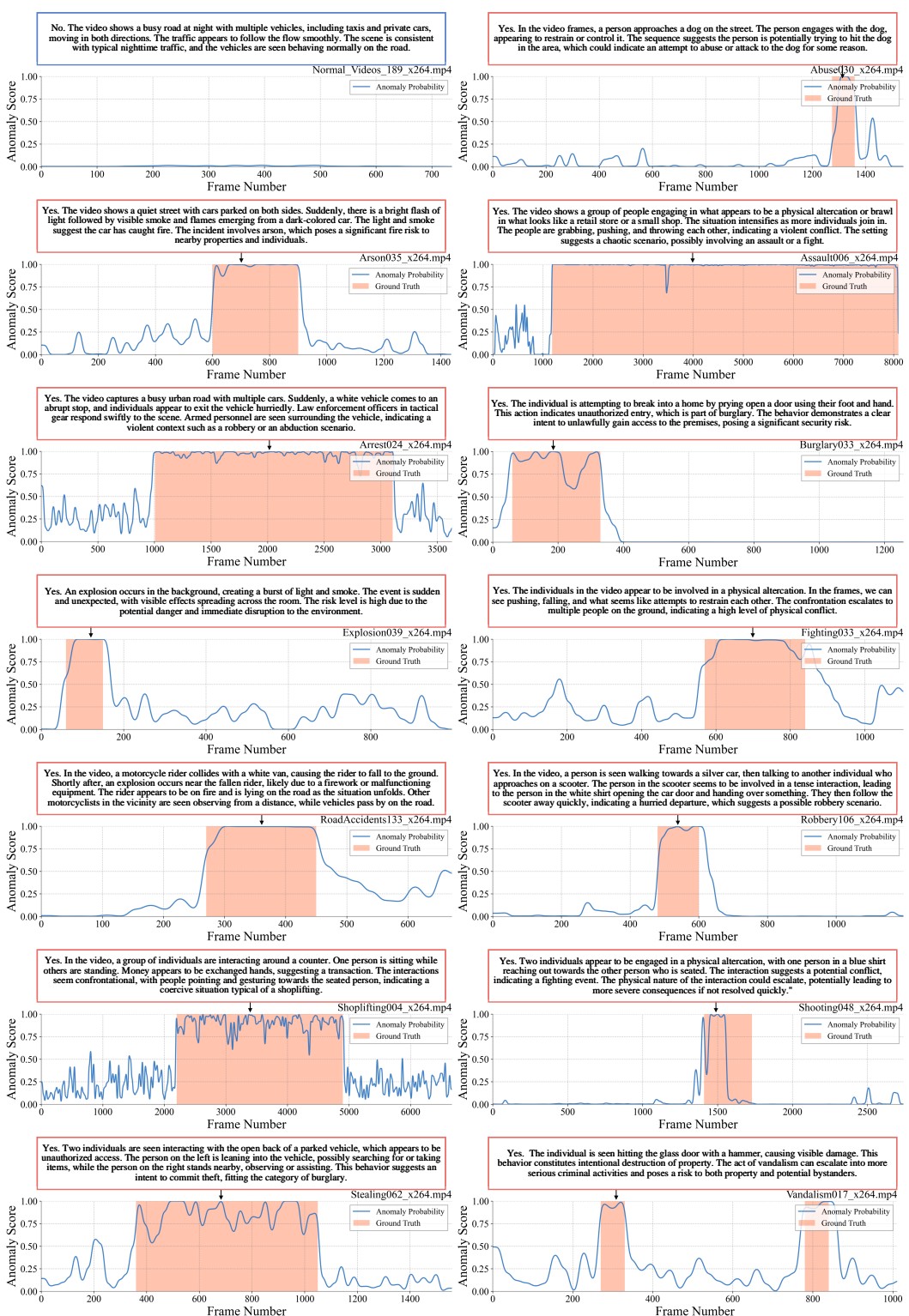

**Figure 12:** Further qualitative results on UCF-Crime dataset. Covered all 13 abnormal category and normal examples.

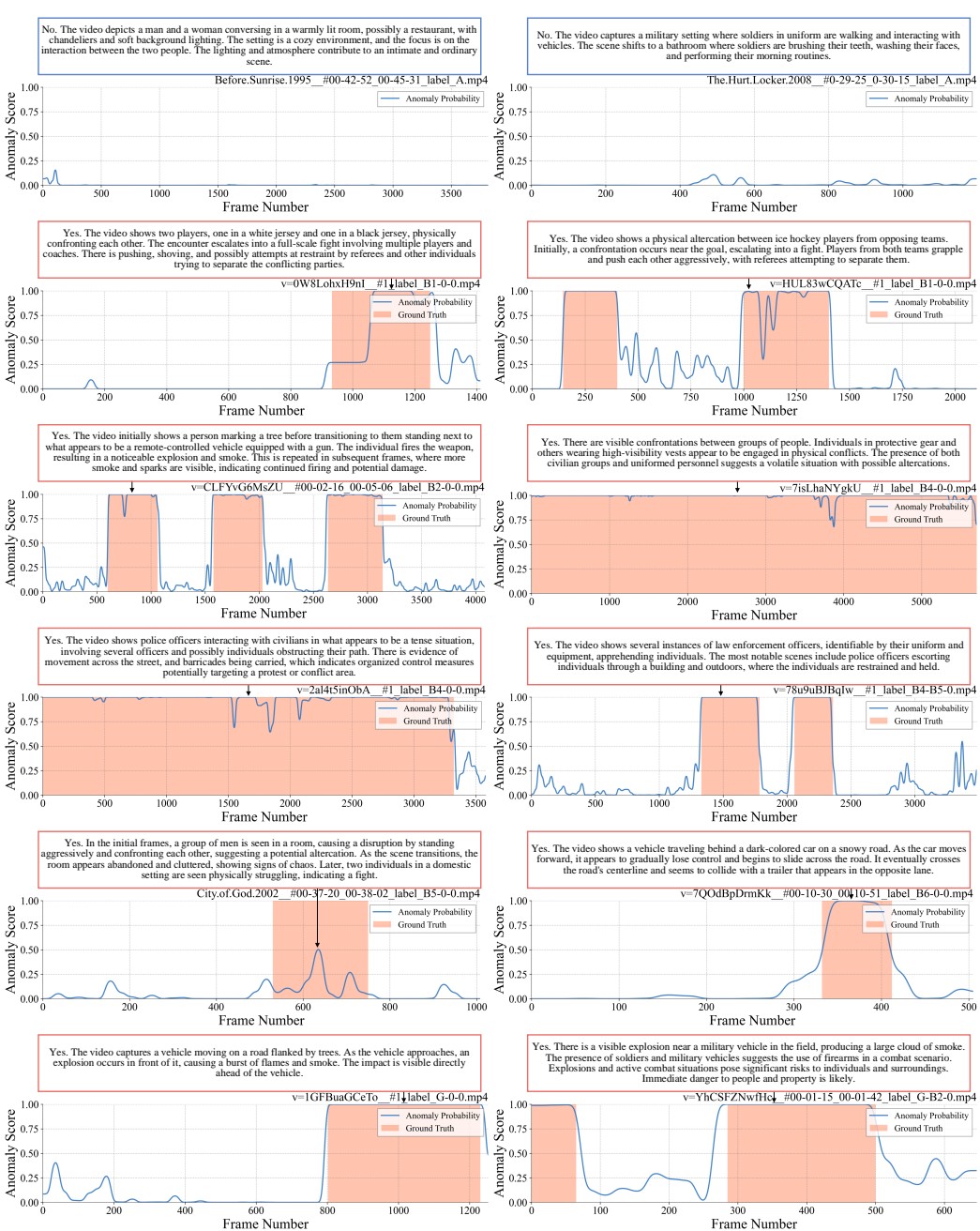

**Figure 13:** Further qualitative results on XD-Violence dataset. Covered all 6 abnormal category and normal examples.

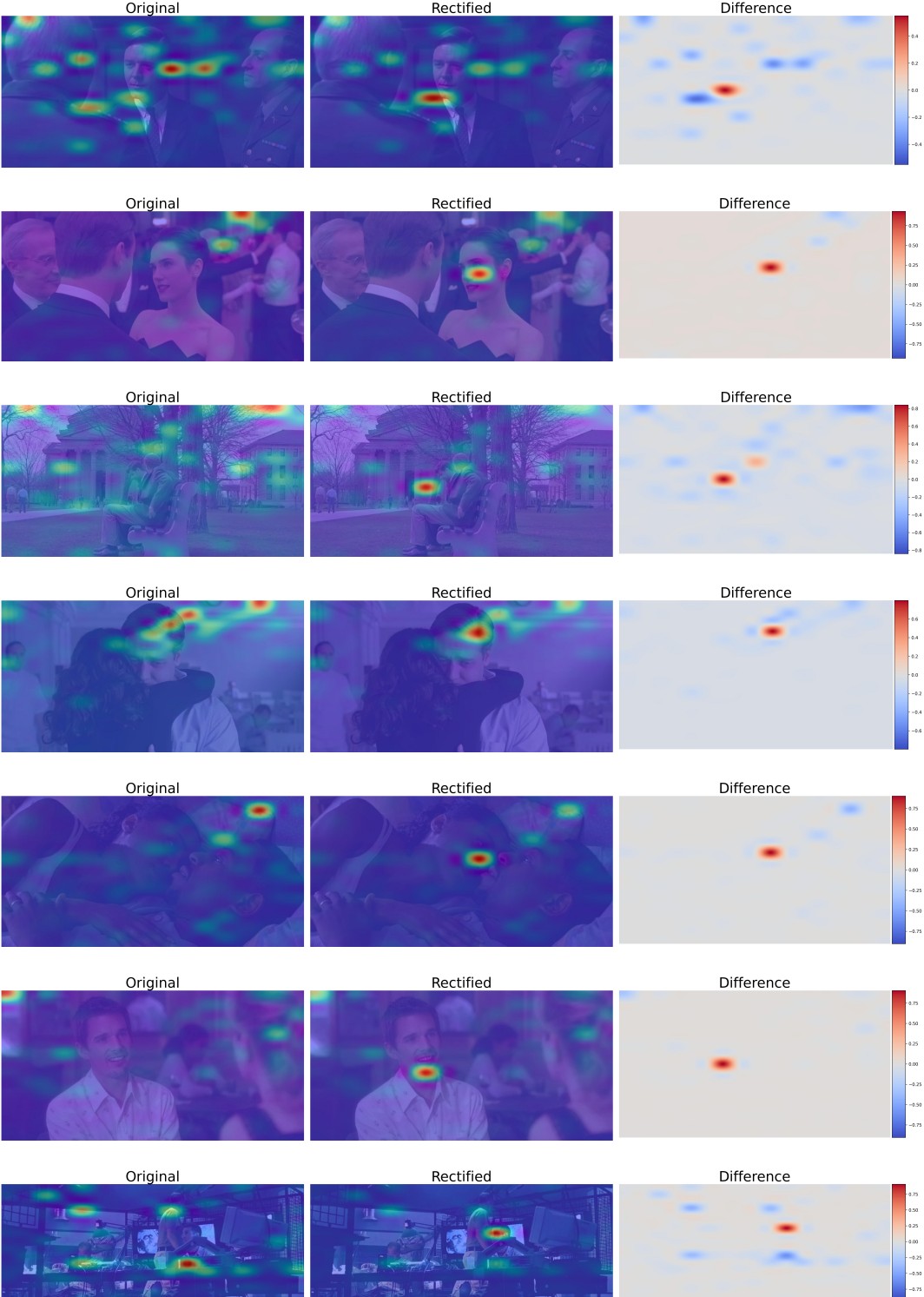

**Figure 14:** Further visualizations of SteerVAD on XD-Violence datasets. Our steering method effectively rectify the biased or ambiguous focus on the original videos.

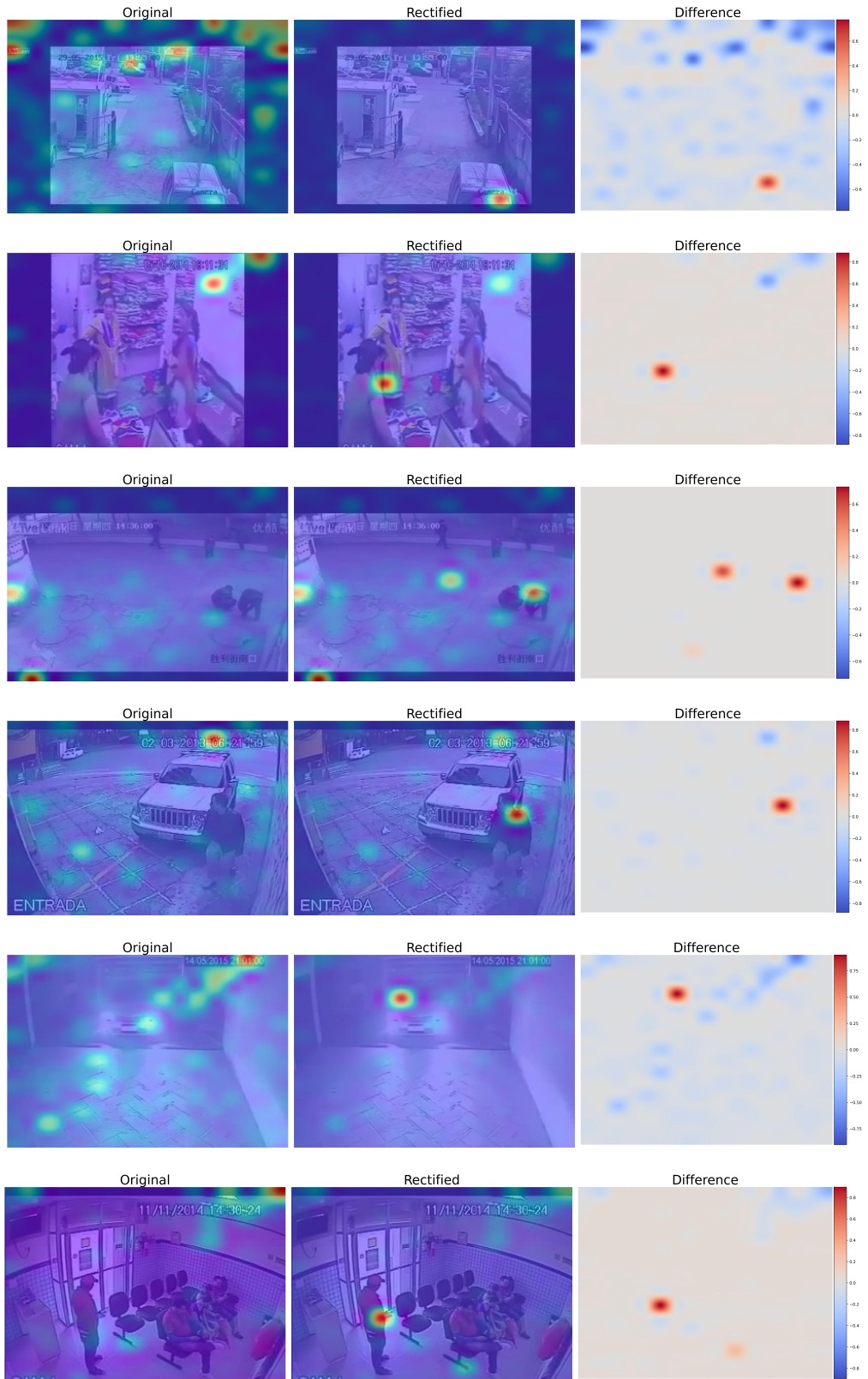

**Figure 15:** Further visualizations of SteerVAD on UCF-Crime datasets. Our steering method effectively rectify the biased or ambiguous focus on the original videos.

