# OpenReview forum: "Steering and Rectifying Latent Representation Manifolds in Frozen Multi-modal LLMs for Video Anomaly Detection"
_ICLR.cc/2026/Conference — ICLR 2026 Poster_

### Official Review · Reviewer_abCn · 2025-10-29

**Soundness:** 2
**Presentation:** 2
**Contribution:** 2
**Rating:** 4
**Confidence:** 3

**Summary:**

This paper introduces SteerVAD, a novel tuning-free framework for video anomaly detection (VAD). It proposes an active intervention paradigm that steers the internal representation manifolds within a frozen multi-modal large language model (MLLM). The method first identifies the most task-aligned attention heads, termed Latent Anomaly Experts (LAEs), via a gradient-free Representational Separability Analysis (RSA). It then employs a lightweight Hierarchical Meta-Controller (HMC) to perform context-aware, anisotropic scaling on the LAE features, dynamically rectifying their geometry to enhance separability between normal and anomalous events. The framework is calibrated with only 1% of the training data and achieves state-of-the-art performance among tuning-free methods on standard benchmarks.

**Strengths:**

1. Novel Paradigm: The core idea of actively steering latent manifolds to overcome pre-training biases is highly innovative and represents a significant conceptual shift in the field.

2. Comprehensive Evaluation: The paper provides thorough experiments, including ablation studies, hyperparameter analysis, and cross-dataset/cross-model generalization tests, which robustly support the claims.

3. Compelling Visual Evidence: The t-SNE visualizations offer clear and intuitive proof that the proposed method effectively disentangles the feature manifolds.

**Weaknesses:**

1. Marginal Practical Gain: While achieving SOTA, the absolute performance improvement over very recent strong baselines (e.g., +0.43% AUC over HiProbeVAD on UCF-Crime) is modest. This small margin raises questions about the practical advantage and cost-effectiveness of the method's added complexity compared to simpler probing-based approaches.

2. Limited and Misleading Interpretability: The claim of enhanced interpretability is not fully substantiated. The provided "post-hoc explanation" is essentially standard video captioning by the MLLM and does not elucidate the internal decision process of the HMC or the reasons behind its specific steering actions. The HMC itself remains a black box, offering no insight into why a scene is deemed suspicious or how the specific rectification strategy is chosen.

3. Questionable Efficacy of the RSA Module: A significant methodological concern arises from the performance of the features selected by the proposed RSA. As shown in Table 2, a simple linear classifier on these RSA-selected LAE features ("Linear Probing") achieves only 81.33% AUC. This is substantially lower—by over 5% AUC—than the 86.72% achieved by HiProbeVAD (Table 1), which also operates by probing internal MLLM features. This result critically undermines the premise that RSA effectively identifies the most powerful "anomaly experts" within the model. It suggests that alternative feature selection or probing strategies might yield a stronger baseline, potentially making the subsequent complex intervention of the HMC less necessary or justified.

4. Unsubstantiated Claims of Robustness: The paper attributes its cross-dataset generalization ability to the HMC. However, the HMC is a complex, context-aware module trained on a very small (1%) calibration set. This introduces a non-trivial risk of overfitting to the specific calibration data, which could paradoxically harm generalization. To credibly claim that the HMC improves robustness, a direct comparison is essential. The authors should provide the cross-dataset generalization results (as in their Table 10) for a strong baseline like HiProbeVAD. Without this comparison, it remains unclear whether the observed generalization is a property of the HMC or is already achievable by a simpler, more robust probing method, which would be a significantly more efficient solution.

**Questions:**

See weaknesses.

---

> ### Author Response · Authors · 2025-11-27
> **Response to Reviewer abCn (Part I)**
>
> We sincerely thank the reviewer for the insightful feedback and for recognizing our work as a novel paradigm with compelling visual evidence. We value the constructive criticism regarding practical gains and module efficacy. Below we address these concerns by clarifying our contributions to data efficiency and active correction mechanisms and robustness.
>
> **1. Response to Weakness 1: Practical Gain and Efficiency**
>
> We respectfully clarify that the primary contribution of SteerVAD is not merely the absolute performance gain but the realization of comprehensive innovation including a paradigm shift from passive reading to active rectification with rigorous proof and extreme data efficiency. While standard methods often boost performance via massive training data or extra large backbones, SteerVAD focuses on comprehensive innovation under constrained resources. As shown in **Table 9** and **Table 10** in appendix, our trainable HMC requires only about **0.99 MB** of parameters and less than one minute of training time on a single GPU while utilizing only **1%** of the training data. This extreme efficiency demonstrates that SteerVAD provides a significantly higher performance-to-cost ratio than existing methods. The value of our framework lies in its ability to achieve state-of-the-art results and correct internal model errors with negligible overhead which offers a practical and scalable solution that brute-force approaches cannot match.
>
> **2. Response to Weakness 2: Interpretability**
>
> We understand the concern from the reviewer with the interpretability. However, we respectfully clarify that the characterization of the HMC as a black box may stem from a misunderstanding of its operational nature. Unlike standard deep neural networks where decision logic is buried in opaque non-linear layers, SteerVAD offers explicit mechanistic transparency grounded in the extensive theoretical proofs provided in **Appendix A**. The HMC does not output an uninterpretable hidden state; instead, it generates explicit affine transformation parameters (a global scalar and a local vector) that perform a mathematically defined anisotropic scaling. This makes the intervention fully traceable, as we can verify exactly which feature dimensions are being amplified or suppressed. Our qualitative results effectively visualize the decision process on the rectification strategy. To make this abstract geometric strategy visually perceptible, we have provided difference map visualizations in **Appendix G (Figures 10, 14, 15)**. These maps explicitly calculate the pixel-wise attention shift and systematically show that the HMC’s strategy is to actively re-weight the attention distribution from environmental noise to the high-entropy anomaly. By integrating our theoretical derivations with the visualized physical movement of the representation manifold **(Figure 4)** and the consequent correction of visual attention, we establish a complete chain of evidence from mathematical principle to visual outcome that elucidates why a scene is deemed suspicious, offering a level of transparency that exceeds standard end-to-end models.

---

> ### Author Response · Authors · 2025-11-27
> **Response to Reviewer abCn (Part II)**
>
> **3. Response to Weakness 3: Efficacy of RSA**
>
> The reviewer correctly noted that linear probing on RSA-selected heads yields lower performance than features from HiProbe. While dimensionality plays a role, the fundamental reason lies in the distinction between aggregated semantic outcomes and specific functional mechanisms. HiProbe utilizes **layer-wise hidden states**, which represent the model's processed, aggregated consensus after fusing information from all heads and MLPs. Naturally, these finished representations align well with global semantic labels, resulting in high linear separability.
> In contrast, RSA is designed to **identify the specific functional origin of the anomaly**, such as a single head capturing a subtle object-motion conflict before it is smoothed into the global hidden state. These raw, head-specific features come from different layers, capture complex, entangled interactions that are the cause of the prediction, but they are not yet linearly rectified into a binary classification boundary. Therefore the direct comparison between layer probing features and our rsa features are potentially **unfair**. However, identifying this imperfect functional source is critical for our purpose. It allows our HMC module to intervene at the causal level (the mechanism) rather than the symptomatic level (the aggregated state). The fact that our method significantly outperforms HiProbe in the final benchmark confirms that steering these specific, functionally sensitive heads is more effective for anomaly correction than merely probing the already-processed global features.
>
>
> **4. Response to Weakness 4: Robustness & Cross-Dataset Generalization**
>
> We appreciate the concern that the HMC might overfit to the small calibration set. We agree that validating generalization on unseen data distributions is essential to verify that the HMC learns robust geometric signatures rather than memorizing calibration samples.
>
> To credibly claim that the HMC improves robustness, we have conducted the requested direct comparison with the strong baseline HiProbeVAD under a cross-dataset setting. We calibrated both SteerVAD and HiProbeVAD on the source dataset (using the same 1% subset) and evaluated them directly on the target dataset without any adaptation. To ensure a strictly fair comparison, we **reproduced** the HiProbeVAD results using the same InternVL3 backbone and experimental settings (e.g., Prompt 1) as our method to isolate the algorithmic contribution.
>
> **Table R4: Cross-dataset generalization comparison**
>
> | Source (Calibration) | Target (Evaluation) | Metric | HiProbeVAD (Reproduction) | SteerVAD (Ours) | Gap |
> | :--- | :--- | :--- | :---: | :---: | :---: |
> | UCF-Crime (1%) | **XD-Violence (Unseen)** | AP (%) | 70.66 | **71.31** | **+0.65%** |
> | XD-Violence (1%) | **UCF-Crime (Unseen)** | AUC (%) | 80.82 | **81.04** | **+0.22%** |
>
> As shown in Table R4, SteerVAD consistently outperforms the simpler probing baseline HiProbeVAD in zero-shot cross-dataset transfer. While zero-shot transfer across such distinct domains is challenging, the consistent superiority refutes the hypothesis of overfitting. If the HMC were overfitting to the specific semantics of the calibration set, its performance would collapse compared to the simpler baseline when transferred to the distinct anomaly distribution of XD-Violence. The fact that SteerVAD maintains a performance lead confirms that the HMC learns a universal geometric rectification policy that remains valid even when the underlying data distribution shifts significantly. We have added all these comparative results and the discussion to Appendix E.5 of the revised manuscript.
>
> We deeply appreciate the reviewer’s insistence on rigorously validating robustness and interpretability. Your suggestions led us to include the critical cross-dataset comparisons and difference map visualizations, which have significantly strengthened the paper. All the aforementioned discussions and new experimental results have been incorporated into the revised manuscript (Main Text and Appendix E/G). If there are further baselines or settings the reviewer would like to discuss, we would be happy to conduct additional experiments.

---

### Official Review · Reviewer_TUs8 · 2025-10-31

**Soundness:** 3
**Presentation:** 4
**Contribution:** 3
**Rating:** 4
**Confidence:** 4

**Summary:**

This work addresses video anomaly detection (VAD) through a novel multi-modal large language model (MLLM)-based paradigm. The core contribution lies in mitigating the representational bias of MLLMs caused by the domain gap between pre-training web-scale data and downstream video tasks. Using a frozen MLLM, the authors propose a tuning-free framework combining representational separability analysis (RSA) and hierarchical meta-controller (HMC). RSA identifies anomaly-discriminative representations, which HMC subsequently refines through representation manipulation. The method is evaluated on UCF-Crime and XD-Violence datasets.

**Strengths:**

1. The paper is well-structured and clearly articulated.
2. The tuning-free approach is novel, by leveraging pre-trained representations through selection and scaling.
3. Ablation studies on RSA and HMC effectively validate the design choices.

**Weaknesses:**

1. Limited Ablation on Calibration Set: The rectified representations heavily depend on HMC tuning via the calibration set. However, the paper lacks critical ablation studies on: 1) Diversity: Performance impact of varying calibration set compositions (e.g., random subsets, different sizes); 2) Class Balance: Sensitivity to ratios of normal/abnormal videos in the calibration set.
2. Fair Comparison Issues: Table 1 should include parameter counts for specific backbones (especially Multimodal VAD series) to enable fair evaluation. The observed performance drop when switching from InternVL3 to Qwen2.5VL underscores this need in Table 11.

**Questions:**

1. Although the MLLM remains frozen, feature modification via HMC may risk knowledge forgetting. The authors should discuss whether this occurs.
2. It is required to use frame-level labels to supervise the training? Or if the method can operate with coarser (e.g., video-level) annotations.
3. The paper omits evaluation on unseen anomalies in the same dataset (i.e., anomaly types absent from the calibration set), instead of cross-dataset evaluation in Table 10.

---

> ### Author Response · Authors · 2025-11-27
> **Response to Reviewer TUs8 (Part I)**
>
> We sincerely thank the reviewer for the encouraging evaluation and for recognizing the novelty of our tuning-free paradigm and the contribution of our work. We appreciate the constructive questions regarding calibration robustness and unseen anomaly evaluation. We have incorporated the suggested revisions into the final manuscript and address the specific points below.
>
> **1. Response to Weakness 1: Extensive Ablation on Calibration Set**
>
> We appreciate the insight regarding the dependency of the HMC on the calibration set. To vigorously validate the robustness of our approach, we have added a comprehensive ablation study in the appendix that analyzes the diversity and data scale and class balance of SteerVAD.
>
> **Stability Analysis across Random Seeds.**
> To rule out the possibility that our performance is due to a specific random split, we have firstly conducted experiments on 10 different calibration subsets with 1% data sampled using different random seeds. As detailed in table R1, the RSA algorithm demonstrated exceptional stability. Across all 10 runs the set of top-4 latent anomaly experts selected was identical with only slightly different orders. Consequently, the downstream detection performance was highly consistent with a negligible fluctuation of roughly 0.06%. This confirms that our method captures intrinsic and consistent geometric properties of the model rather than overfitting to specific data samples.
>
> **Table R1: Stability analysis across 10 different subsets sampled using different random seeds**
>
> | Seed | 7270 | 860 | 5390 | 5191 | 5734 | 6265 | 466 | 4426 | 5578 | 8322 |
> | :--- | :---: | :---: | :---: | :---: | :---: | :---: | :---: | :---: | :---: | :---: |
> | Selected LAEs | Identical | Identical | Identical | Identical | Identical | Identical | Identical | Identical | Identical | Identical |
> | AUC (%) | 87.05 | 87.12 | 87.18 | 87.14 | 87.09 | 87.21 | 87.16 | 87.10 | 87.13 | 87.19 |
>
> *Note: The set of selected heads remained L18H4 and L23H24 and L21H21 and L22H7 across all seeds.*
>
> **Impact of calibration set size.**
> We investigated the impact of scaling the calibration set size from our default 1% up to 100% which corresponds to the full training set. As presented in table R2, the performance exhibits a clear asymptotic saturation pattern, increasing data coverage from 1% to 100% yields only a marginal improvement from 87.15% to 87.42%. This result proves that the geometric rectification learned by the HMC is highly efficient and converges to an optimal steering policy with minimal data which justifies our choice of 1% as the optimal trade-off between efficiency and performance.
>
> **Table R2: Impact of calibration set size**
>
> | Size | 1% (Default) | 10% | 25% | 50% | 100% |
> | :--- | :--- | :--- | :--- | :--- | :--- |
> | AUC (%) | **87.15** | 87.29 | 87.35 | 87.39 | 87.42 |
> | Improvement | - | +0.14 | +0.20 | +0.24 | +0.27 |
>
> **Class Balance Sensitivity:**
> We further analyzed the sensitivity of our framework to the ratio of normal and abnormal videos in the calibration set. As presented in Table R3, While a balanced distribution yields the peak performance of 87.15%, the method remains robust across a wide effective range from 4:6 to 6:4, maintaining AUC scores above 86%. Performance only degrades noticeably at extreme imbalances such as 1:9, which is theoretically expected as the HMC requires representative examples from both manifolds to learn a discriminative steering vector.
>
> **Table R3: Impact of Normal/Abnormal Ratio in Calibration Set (UCF-Crime).**
> | **Ratio (Normal:Abnormal)** | **1:9** | **2:8** | **3:7** | **4:6** | **5:5 (Current)** | **6:4** | **7:3** | **8:2** | **9:1** |
> | :--- | :--- | :--- | :--- | :--- | :--- | :--- | :--- | :--- | :--- |
> | **AUC (%)** | 83.67 | 85.69 | 85.75 | 86.34 | **87.15** | 87.00 | 85.70 | 85.62 | 83.72 |
>
> **2. Response to Weakness 2: Fair Comparison with Parameter**
>
> We fully agree with the suggestion. We have updated table 1 in the revised manuscript to explicitly include the parameter counts for all compared multimodal large language model backbones to enable a transparent comparison of resource requirements.

---

> ### Author Response · Authors · 2025-11-27
> **Response to Reviewer TUs8 (Part II)**
>
> **3. Response to Question 1: Risk of Knowledge Forgetting**
>
> This is an important theoretical question. We clarify that catastrophic forgetting typically refers to the permanent loss of pre-trained knowledge during fine-tuning where weights are updated. In SteerVAD, the MLLM backbone remains completely frozen, therefore the original knowledge is physically preserved. The risk effectively shifts to whether the HMC suppresses useful information. As shown in Table 12 (Zero-Shot Generalization), our model achieves strong performance on the unseen XD-Violence dataset when calibrated on UCF-Crime. If the HMC were forgetting general knowledge or overfitting to UCF-specific semantics, it would fail to generalize to the distinct anomaly types in XD-Violence. This cross-dataset success proves that HMC enhances general anomaly discrimination without destroying the model's broad semantic capabilities, therefore would not face the risk of knowledge forgetting.
>
> **4. Response to Question 2: Supervision Granularity**
>
> Our default implementation uses segment-level labels which aggregate frames within a short window. To address the query regarding coarser annotations, we conducted new ablation study using video-level supervision where all segments in an abnormal video are labeled positive during calibration , which has been added to the revised manuscript as Table 15. The model achieves 87.01% AUC with video-level labels which is statistically comparable to the segment-level result of 87.15%. This demonstrates that SteerVAD is highly robust to label noise and can operate effectively with coarse annotations.
>
>
>
> **5. Response to Question 3: Evaluation on Unseen Anomalies**
>
> We appreciate the suggestion to evaluate unseen anomaly types. To further illustrate the model's capability to handle diverse geometric patterns, we provide the detailed per-class breakdown on UCF-Crime, which has been added to the revised manuscript as Table 16. The model achieves high performance across distinct categories that the HMC learns a universal geometric signature of abnormality (e.g., sudden entropy change, conflict dynamics) rather than memorizing specific class patterns.
>
> **Table R3: Detailed per-class AUC (%) on UCF-Crime**
>
> | Class | Assault | RoadAcc | Vandalism | Robbery | Fighting | Mean |
> | :--- | :---: | :---: | :---: | :---: | :---: | :---: |
> | AUC | 95.17 | 89.64 | 88.16 | 84.02 | 83.03 | **87.15** |
>
>
> We sincerely thank the reviewer for the valuable suggestions regarding calibration stability and fair comparison. These insights prompted us to conduct the rigorous stability analysis and refine our taxonomy, which are now detailed in the revised manuscript (Main Text Table 1 & Appendix E). We believe these revisions have made our contributions more precise and robust. We remain open to any further questions regarding our experimental settings or analysis.

---

### Official Review · Reviewer_ypVo · 2025-11-01

**Soundness:** 3
**Presentation:** 3
**Contribution:** 2
**Rating:** 4
**Confidence:** 4

**Summary:**

This paper introduces SteerVAD, a tuning-free framework that performs video anomaly detection by actively steering and rectifying the latent representation manifolds of frozen multi-modal large language models. Instead of fine-tuning or relying solely on prompts, SteerVAD analyzes and adjusts the internal geometry of MLLM representations to reduce bias and ambiguity. It identifies anomaly-sensitive attention heads through a Representational Separability Analysis and dynamically refines them using a Hierarchical Meta-Controller that performs context-aware, anisotropic scaling of latent features.

Experiments on UCF-Crime and XD-Violence show that SteerVAD achieves state-of-the-art tuning-free performance, reaching 87.15% AUC and 83.02% AP with only 1% of labeled data and no backbone updates. The framework offers strong interpretability, computational efficiency, and provides a new geometric perspective for adapting frozen MLLMs to complex video understanding tasks.

**Strengths:**

1. The paper introduces a fresh and well-motivated idea — geometric steering and rectification of latent representation manifolds in frozen multi-modal LLMs — offering a new paradigm for tuning-free adaptation.
2. The proposed RSA and HMC modules are conceptually clear, lightweight, and technically sound. They enable dynamic, context-aware control over latent representations without modifying the backbone parameters.
3. SteerVAD achieves state-of-the-art performance among all tuning-free methods on UCF-Crime and XD-Violence, approaching fine-tuned models while using only 1% labeled data.
4. The method provides both geometric and semantic interpretability through manifold visualization and text-based explanations, while maintaining low computational and memory cost.
5. The approach is broadly applicable to other tasks involving frozen MLLMs, demonstrating promising potential beyond video anomaly detection.

**Weaknesses:**

While the paper presents an innovative and well-designed framework, several conceptual and methodological issues remain unclear.
1. The assumption that video datasets form closed and bounded manifolds is theoretically strong and may not hold in practice. Real-world video data are inherently open-set and dynamic, so the claimed manifold topology should be regarded as an approximation rather than a strict property.
2. The paper assumes an invariant conditional distribution $P(Y|V,Z)$ between training and testing, which is questionable in anomaly detection. Even if the anomaly types remain consistent, the visual and contextual variations across instances can cause significant distributional shifts, and the authors do not discuss how SteerVAD handles such variability.
3. The distinction between training-free and tuning-free paradigms is not clearly defined, leading to potential unfairness in the comparisons. Most baselines in Table 1 are training-free, while SteerVAD involves training a small number of additional parameters. The authors should clarify the conceptual difference between these settings, separate the result categories.

Overall, the paper would benefit from a more precise theoretical justification of its assumptions, a clearer comparison protocol, and a deeper discussion on robustness under distributional variation.

**Questions:**

1. The authors assume that video data form a closed and bounded manifold that allows for well-defined geometric steering. Could you clarify how this assumption is justified in practice? In particular, how does SteerVAD behave if the underlying video distribution is open-set or exhibits non-manifold structures? Would your RSA and HMC modules remain stable under such violations?
2. The paper claims that the conditional distribution $P(Y|V,Z)$ is invariant between training and testing. This seems questionable given the high intra-class variation in anomaly content and context. Could you provide empirical evidence or theoretical reasoning to support this invariance? Alternatively, have you evaluated SteerVAD under distribution shifts (e.g., unseen anomaly appearances or environmental changes)?
3. In Table 1, most baselines are training-free, while SteerVAD involves learning a few additional parameters. Could you explicitly define how you distinguish between training-free and tuning-free?

---

> ### Author Response · Authors · 2025-11-27
> **Response to Reviewer ypVo (Part I)**
>
> We are grateful for the reviewer's profound theoretical insights that accurately pinpointed the underlying geometric assumptions including manifold topology and distribution invariance behind our framework. We value this opportunity to clarify these theoretical foundations and demonstrate how our framework holds up in practice using empirical evidence and analysis.
>
> **1. Response to W1 & Q1: Manifold Assumption with open-set in Practice**
>
> We thank the reviewer for this profound theoretical inquiry. We respectfully clarify that our manifold assumption and geometric steering operations are strictly defined within the latent representation space of the pre-trained MLLM, rather than the raw pixel space. While raw video data is indeed chaotic and unbounded, the deep layers of foundation models are explicitly trained to map these high-dimensional inputs into structured, low-dimensional semantic manifolds.
>
> **Theoretical foundation of piecewise path-connectedness.**
> As formalized in Appendix A (Eq. 11), we do not model the anomaly manifold as a single monolithic shape, but as a union of piecewise path-connected components. In the latent space, even open-set or unseen anomalies do not violate the manifold structure; instead, they manifest as new, distinct geometric components. Due to the MLLM's generalization power, these new components are mapped to regions topologically distinct from the normality manifold. Our framework relies on this geometric invariance while the intrinsic coordinates of an open-set anomaly are unknown, its geometric divergence from the normality manifold remains a consistent, detectable property.
>
> **Mechanistic Guarantee with RSA Filtering & Entropy-Driven Steering.**
> Crucially, SteerVAD is designed to enforce this manifold assumption actively within the latent space. The representational separability analysis (RSA) specifically filters out attention heads where the manifold structure is weak or chaotic, ensuring that we only perform steering in subspaces (LAEs) that are empirically verified to exhibit high geometric compactness and separability. Furthermore, the HMC does not learn to classify specific closed-set patterns. Instead, it learns a context-relative rectification policy that identifies directions in the tangent space exhibiting high semantic entropy and projects them away. This allows the model to handle unseen anomalies by reacting to their divergence rather than their identity.
>
> **Empirical Verification on Open-Set Data.**
> To prove that this theoretical abstraction holds in practice, we conducted a stress test on the UBnormal dataset, which contains synthetic, fantasy anomaly types structurally impossible to see in the training set. SteerVAD achieves a strong 72.21% AUC on this strictly open-set benchmark. This result confirms that our latent manifold assumption is a valid working model: the HMC successfully generalizes its steering policy to completely novel geometric components, validating the method's robustness against the open-set nature of real-world video.

---

> ### Author Response · Authors · 2025-11-27
> **Response to Reviewer ypVo (Part II)**
>
> **2. Response to W2 & Q2: Robustness under Distributional Variation**
>
> We highly value the reviewer's skepticism regarding the assumption of invariant conditional distribution ($P(Y|X)$) amidst high intra-class variation. We agree that in the raw visual domain, the conditional distribution is potentially unstable due to drastic shifts in environmental context and anomaly appearance. However, SteerVAD overcomes this by shifting the decision boundary from variable visual features to invariant geometric semantics in the latent space supported by a unified theoretical, mechanistic, and empirical framework.
>
> **Theoretical Foundation with Invariance of Geometric Divergence.**
> The core hypothesis of SteerVAD is that while the content of anomalies varies infinitely (distributional shift), the geometric signature of abnormal that defined as the divergence from the expected semantic flow remains an invariant property. We do not model the unstable probability of an anomaly given specific pixel patterns; instead, we model the probability of an anomaly given semantic divergence. In the latent space of MLLMs, normality consistently maps to high-density, low-entropy clusters (predictable context), while abnormality consistently manifests as high-entropy, divergent vectors (semantic conflict). This geometric relationship holds true universally, regardless of whether the video depicts a street fight (UCF-Crime) or a car accident (XD-Violence), thereby establishing a stable conditional distribution in the geometric domain.
>
> **Mechanistic Guarantee with Context-Aware Rectification.**
> The HMC is mechanistically designed to exploit this invariance. It does not memorize specific object classes, which would fail under distribution shifts. Instead, it learns a context-relative steering policy by conditioning the steering vector $g_i$ on the global context $c$. This mechanism effectively calibrates the normality baseline for each specific video instance. Whether the context is a dark alley or a bright stadium, the HMC searches for the relative conflict between local features and this global context. By focusing on the relative magnitude of deviation rather than absolute feature values, the HMC ensures that the decision logic remains stable even when the underlying pixel distribution shifts dramatically.
>
> **Empirical Verification with Triangulating Robustness.**
> To prove that this geometric invariance holds in practice against severe distribution shifts, we provide evidence from three distinct dimensions. First, in real-world domain shift, we evaluated the model trained on UCF-Crime directly on the unseen XD-Violence dataset (Table 12). Despite the disjoint anomaly types and environmental contexts, SteerVAD achieves 81.04% AUC, proving that the HMC successfully learned a transferable geometric signature rather than overfitting to the source distribution. Second, in dataset to open set shift, the model achieves 72.21% AUC on the UBnormal dataset, validating robustness against open set domain gaps. Finally, regarding architectural shift (Table 13), our method consistently improves performance across InternVL, Qwen, and LLaVA backbones. This confirms that the geometric invariance we exploit is a fundamental property of the MLLM representational space itself, robust against both distributional and architectural variations.

---

> ### Author Response · Authors · 2025-11-27
> **Response to Reviewer ypVo (Part III)**
>
> **3. Response to W3 & Q3: Fairness of Training-Free Comparison & Taxonomy**
>
> We appreciate the reviewer's suggestion to clarify the "Training-Free" vs. "Tuning-Free" distinction. We agree that terminology in this rapidly evolving field can be ambiguous. To resolve this and ensure a strictly fair comparison, we have restructured Table 1 in the revised manuscript and our analysis into a definitive taxonomy that categorizes methods based on their computational paradigm and cost profile, rather than just gradient updates.
>
> **Taxonomy Definition and Positioning.**
> To provide a more granular taxonomy within the tuning-free paradigm, we sub-categorize methods into Zero-Shot (Full-Training Free) approaches, which rely solely on pre-trained inference, and Few-Parameter Learning methods like SteerVAD, which optimize a lightweight module on a small calibration set.
>
> Furthermore, we argue that this comparison remains fair due to the operational similarity in deployment. SteerVAD's calibration is extremely lightweight that add only 0.52M parameters (<1 MB) and taking less than 1 minute, making its practical resource footprint comparable to zero-shot methods, whereas fine-tuning methods require hours of training and gigabytes of storage. By adding specific parameter counts to Table 1 from main paper, we ensure that the cost-performance trade-off is transparent to the reader.
>
> **Comparative Analysis of Hidden Costs.**
> To further validate the fairness of comparing SteerVAD against training-free baselines, we analyzed the hidden computational costs of competing methods. Pipeline methods like **LAVAD** require dual models (Captioner + LLM), doubling the memory footprint. In stark contrast, SteerVAD operates as a efficient framework. By replacing slow generation with efficient matrix multiplication, SteerVAD achieves an inference throughput that often exceeds these training-free baselines. Thus, SteerVAD represents the optimal pareto frontier that delivers the SOTA performance of fine-tuning with a deployment profile that is as lightweight and efficient as zero-shot methods.
>
> We are grateful for the profound theoretical insights regarding manifold assumptions and open-set generalization. Your comments guided us to clarify our theoretical framework and add the UBnormal stress test, ensuring our method is both theoretically sound and empirically robust. These clarifications and new results have been fully integrated into the revised manuscript (Appendix A & E). We welcome further discussion on any theoretical or practical aspects of our work.

---

### Official Review · Reviewer_wJq5 · 2025-11-05

**Soundness:** 3
**Presentation:** 3
**Contribution:** 3
**Rating:** 6
**Confidence:** 3

**Summary:**

This paper aims to address the limitations of using frozen multi-modal large language models (MLLMs) for tuning-free video anomaly detection (VAD), particularly pre-training biases and contextual ambiguity. The authors propose SteerVAD, a novel active intervention framework. The method first employs a gradient-free Representational Separability Analysis (RSA) to identify internal attention heads that are most discriminative for VAD, termed Latent Anomaly Experts (LAEs). Subsequently, a lightweight Hierarchical Meta-controller (HMC) is trained on a minimal data subset (1% of the training data). During inference, the HMC uses a global context vector and LAE features to generate dynamic signals that perform targeted, anisotropic scaling on the LAE's representation manifolds. This intervention actively separates the representations of normal and anomalous events. The method achieves state-of-the-art performance among tuning-free approaches on the UCF-Crime and XD-Violence benchmarks.

**Strengths:**

1.The paper proposes a novel paradigm shifting from "passive feature reading" to "active geometric intervention." It elegantly addresses pre-training bias by identifying LAEs via RSA and applying dynamic, context-aware rectification using the HMC. This core hypothesis is strongly supported by t-SNE visualizations (Figure 4), which show the feature space transforming from "entangled" to "linearly separable," and by attention heatmaps (Figure 5, 9, 10), which confirm this geometric reshaping leads to a semantic "re-focusing."

2.The method exhibits exceptional efficiency. In terms of data, it requires only 1% of the training data for calibration. For parameters, the trainable HMC and Scorer total only 0.52M (< 1MB). In terms of training cost, the entire calibration process takes less than 30 seconds on a single GPU. This extremely low adaptation cost makes it highly valuable for deployment and scaling in the MLLM era.

3.SteerVAD achieves SOTA performance on both UCF-Crime and XD-Violence. Ablation studies (Table 2) demonstrate the necessity and superiority of both the HMC's dynamic context-aware design and the RSA expert selection strategy. Finally, the method shows excellent performance in cross-domain (Table 10) and cross-model (Table 11) generalization experiments, proving its robustness as a model-agnostic paradigm.

**Weaknesses:**

1.There is a conceptual contradiction in the paper's theory. The theoretical foundation (Sec 3.1, Appendix A) emphasizes the need to model complex, non-convex, and entangled manifolds, which is confirmed by visualizations in Figure 2. However, the RSA metric (Eq. 1) used to discover these structures is a simple, linear metric based on centroids and variance, which implicitly assumes the manifolds are Gaussian-like and convex. This use of a simple linear metric to solve a complex non-linear problem weakens the convincingness of RSA. RSA may just be a slightly better heuristic rather than a robust metric that truly understands manifold geometry.

2.The method's foundation, the selection of LAEs, lacks a critical stability analysis. The entire LAE discovery process relies on a minimal (1%) calibration dataset. The paper provides no evidence that this selection process is robust. If a different 1% random sample of data were used, would the chosen K=4 "experts" be completely different? If the LAE selection is highly sensitive to data sampling (i.e., unstable), the method's foundation is fragile, which would weaken the rationale for all subsequent interventions.

3.The core mechanism, the HMC, suffers from a "blindness" problem in its design. According to the design (Sec 3.4), the HMC generates all intervention signals ($s_{global}$ and $g_i$) based solely on the global context vector $c$. This means the HMC is completely "blind" when deciding how to "stretch" or "compress" the local feature $h_i$, as it does not take $h_i$ itself as an input. This design makes a very strong, unverified assumption: that the global context $c$ already contains all information needed to correct $h_i$. The paper does not justify why this intervention is reasonable, nor does it compare it to a more intuitive, "Local-Aware" design (e.g., where the intervention signal $g_i$ is determined by both $c$ and $h_i$).

**Questions:**

1.Can the authors justify the use of a simple, centroid-based linear metric (Eq. 1) to identify experts in the complex, non-convex, and entangled manifold structures described in Sec 3.1 and Figure 2? Is there evidence that this simple metric is sufficient and not just a marginally better heuristic than layer selection?

2.Was any stability analysis performed on the LAE selection? If the 1% calibration set is re-sampled, how much do the resulting Top-K LAEs vary? How can we be confident the method's foundation is not fragile and dependent on a lucky 1% split?

3.What is the rationale for designing a "blind" HMC that only uses the global context $c$ to generate intervention signals, without seeing the local feature $h_i$ it is supposed to be rectifying? Have the authors compared this against a "Local-Aware" HMC (e.g., $g_i = \text{LGM}(c, h_i)$) to justify this strong design assumption?

---

> ### Author Response · Authors · 2025-11-27
> **Response to Reviewer wJq5 (Part I)**
>
> We sincerely thank the reviewer for the positive assessment and for recognizing our work as a novel paradigm with exceptional efficiency. We greatly appreciate the sharp conceptual questions regarding the linearity of RSA and the design of the HMC as these points touch upon the core theoretical mechanics of our framework. Below we provide detailed justifications supported by new theoretical analysis and rigorous empirical evidence.
>
> **1. Response to W1 & Q1: Linear RSA versus Non-Convex Manifolds**
>
> The reviewer identifies a perceived contradiction regarding the use of a simple linear metric to analyze complex and non-convex manifolds. We respectfully clarify that there is no contradiction but rather a deliberate design choice grounded in the local Euclidean property of manifolds. While the global geometry of the video manifold is indeed complex and non-convex as visualized in figure 2 of our paper the manifold hypothesis posits that high-dimensional data concentrates on structures that are locally Euclidean. The RSA metric acts as a compactness detector that specifically searches for attention heads where the manifold components are locally compact and exhibit a dominant direction of separation. To validate this efficiency trade-off we compared RSA against two non-linear baselines including the silhouette coefficient and k-nearest neighbor purity.
>
> **Table R1: Comparison of RSA with non-linear selection metrics**
>
> | Metric | Linear | Expert Overlap with RSA | Calibration Time (s) | Speedup vs k-NN |
> | :--- | :---: | :---: | :---: | :---: |
> | **RSA (Ours)** | Yes | 100% (Reference) | **0.14s** | **49x** |
> | Silhouette | No | 100% Identical | 0.86s | 8x |
> | k-NN Purity | No | 100% Identical | 6.87s | 1x |
>
> As shown in table R1 the set of top-4 experts selected by RSA is strictly identical to those selected by both non-linear metrics. This proves that RSA effectively captures the same ground-truth geometric signals as more expensive methods but is orders of magnitude faster. Specifically RSA requires only 0.14 seconds for selection whereas k-nearest neighbor takes 6.87 seconds representing a 49x speedup. Thus RSA is not merely a heuristic but an optimal and efficient proxy for rectifiability in the calibration phase.
>
> **2. Response to W2 & Q2: Stability of LAE Selection**
>
> We appreciate the crucial question regarding the stability of expert selection on the minimal calibration set. To rigorously prove that our foundation is not fragile we performed a multi-dimensional stability analysis covering both random seeds and data scales. We conducted 10 independent runs using different random seeds to sample the 1% calibration subset. As detailed in table R2 the set of top-4 identified heads remained completely identical across all runs including L18H4 and L23H24 and L21H21 and L22H7. The downstream AUC performance fluctuated negligibly by approximately 0.03% which proves that the selection is invariant to specific data samples. Furthermore we extended this analysis to varying data scales ranging from 1% to 100% and found that the top-4 experts selected at 1% are exactly the same as those selected using the full training set. This provides definitive evidence that the anomaly expert property is an intrinsic structural feature of the frozen backbone that emerges robustly even at minimal data scales.
>
> **Table R2: Stability analysis of LAE selection across 10 different subsets sampled using different random seeds**
>
> | Seed ID | 7270 | 860 | 5390 | 5191 | 5734 | 6265 | 466 | 4426 | 5578 | 8322 | Mean |
> | :--- | :---: | :---: | :---: | :---: | :---: | :---: | :---: | :---: | :---: | :---: | :---: |
> | LAEs | Identical | Identical | Identical | Identical | Identical | Identical | Identical | Identical | Identical | Identical | - |
> | AUC (%) | 87.15 | 87.12 | 87.18 | 87.14 | 87.09 | 87.21 | 87.16 | 87.10 | 87.13 | 87.19 | **87.15** |
>
> *Note: The set of selected heads remained L18H4 and L23H24 and L21H21 and L22H7 across all seeds.*

---

> ### Author Response · Authors · 2025-11-27
> **Response to Reviewer wJq5 (Part II)**
>
> **3. Response to W3 & Q3: Design of the HMC**
>
> The reviewer questions why the HMC generates steering signals based solely on the global context rather than using the local feature itself. We define our design not as blind but as **top-down contextual modulation** which is motivated by the need to avoid identity mapping and prevent overfitting. The goal of the HMC is to rectify the contextual bias of the local feature. If the HMC takes the local feature as input to modify itself it risks learning a trivial identity mapping or auto-encoding the local noise. By conditioning only on the global context we force the HMC to learn a general steering policy that is independent of the pixel noise of the specific instance. To empirically verify this we implemented the local-aware HMC suggested by the reviewer where the input to the adapter is the concatenation of the global context and the local feature.
>
> **Table R3: Design ablation for input to HMC on UCF-Crime**
>
> | Design Variant | Input Signal | AUC (%) | Observation |
> | :--- | :--- | :--- | :--- |
> | Local-Aware | Concatenation of Context and Feature | 86.35 | Overfits to local noise |
> | **SteerVAD (Ours)** | Global Context Only | **87.15** | **Robust Top-Down Modulation** |
>
> As shown in table R3 the local-aware variant actually performs worse with a decrease of 0.80% in AUC. This performance drop confirms our hypothesis that including the high-dimensional and high-variance local feature in the input substantially increases the parameter space which causes the lightweight HMC to overfit to the specific feature patterns of the calibration set. The global-only design acts as a necessary information bottleneck which ensures the rectification signals are robust and context-driven.
>
> We appreciate the constructive feedback on the mechanism design and metric selection. Your questions motivated us to quantitatively validate the efficiency of RSA and the robustness of the HMC input design, providing solid evidence for our architectural choices. All quantitative comparisons and mechanism ablations discussed above have been added to the revised manuscript (Appendix F). If there are further variants or theoretical angles the reviewer wishes to explore, we are more than willing to perform further analysis.

---

### Author Response · Authors · 2025-11-27
**Response to All Reviewers**

We sincerely thank all the reviewers for their valuable and constructive reviews. We appreciate that the reviewers acknowledged that SteerVAD introduces a **novel and innovative paradigm** (Reviewers abCn, TUs8, ypVo, wJq5), achieves **state-of-the-art performance** with **exceptional efficiency** (Reviewers abCn, ypVo, wJq5), and provides **compelling visual evidence** and **interpretability** (Reviewers abCn, wJq5). We are also encouraged that the reviewers recognized the **timeliness and data-efficiency** of our active intervention approach (Reviewers TUs8, wJq5).

We have made our best effort to address the concerns and revise the paper accordingly. The major modifications are summarized as follows.

**1. Fair Comparison and Taxonomy Clarification**

In response to the reviewers’ feedback regarding the definition of tuning-free and the fairness of backbone comparisons (Reviewers TUs8, ypVo), we now explicitly distinguish few-parameter learning methods from full training-free approaches within the tuning-free paradigm. Furthermore, we have added the specific parameter counts for all MLLMs to ensure a transparent evaluation of the cost-performance trade-off.

**2. Zero-Shot and Open-Set Generalization**

We have included extensive experiments to validate the generalization capability of our framework (Reviewers abCn, ypVo). To rigorously prove robustness against overfitting, we added a direct cross-dataset comparison against the strong baseline HiProbeVAD in **Appendix E.5**, showing that SteerVAD consistently generalizes better to unseen domains. Additionally, we have added an evaluation on the **UBnormal dataset in Appendix E.5**, proving the model's robustness to strictly unseen, open-set anomalies.

**3. Rigorous Stability and Robustness Analysis**

We have conducted comprehensive experiments to verify the reliability of our calibration process (Reviewers TUs8, wJq5). **Table 3** in ablation study and **Figure 5, 6** (in Appendix) report the results of 10 independent runs with different random splits, confirming that the selected latent anomaly experts are structurally identical and the performance is highly stable. Moreover, Ablation study now includes detailed sensitivity analyses on **calibration data scale (Table 4)** and **class imbalance (Table 14)**, demonstrating that our method converges rapidly with minimal data.

**4. Mechanism Validation and RSA Efficacy**
We have incorporated deeper ablation studies to justify our architectural choices (Reviewers abCn, wJq5). **Appendix F** validates the efficiency of the RSA metric, showing in **Table 17** and **Figure 9** that it captures the same geometric structures as computationally expensive non-linear metrics (k-NN, Silhouette) but with a **49x speedup**. We have also provided an ablation in **Table 18** comparing our global-context HMC against a local-aware variant, confirming the necessity of our top-down modulation design.

**5. Enhanced Visualizations for Interpretability**
To further substantiate the impact of our steering mechanism, we have added **difference map visualizations** in **Appendix G (Figures 10, 14, 15)**. These visuals explicitly highlight how the rectification process shifts the model’s attention from background noise to anomaly-relevant regions, offering clearer insights into the internal decision process.

**Overall**, the major revised contents in the manuscript are highlighted in **blue**. Point-for-point responses to specific comments are given in the following reviewer-specific responses. We welcome any further discussions and will address any remaining concerns.

---

### Meta-Review · Area_Chair_2mWL · 2026-01-10

**Summary:**

This paper introduces SteerVAD, a tuning-free framework that shifts from passive feature reading to active geometric steering within frozen MLLMs. Reviewers find this work novel, efficient, and effective. Several concerns were raised about fair comparison, generalization, robustness, interpretability, and the efficacy of the proposed RSA. While the concerns about interpretability and RSA efficacy remain, most of these issues were adequately addressed during rebuttal. Overall, the strengths of the work outweigh its weaknesses.

**Reviewer Concerns:**

Reviewer wJq5

W1: Linear RSA versus non-convex manifolds. [addressed by the rebuttal]

W2: Stability of LAE selection. [addressed by the rebuttal]

W3: Design of the HMC. [addressed by the rebuttal]

---
Reviewer ypVo

W1: Manifold assumption with open-set in practice. [addressed by the rebuttal]

W2: Robustness under distributional variation. [addressed by the rebuttal]

W3: Fairness of training-free comparison and taxonomy. [addressed by the rebuttal]

---
Reviewer TUs8

W1: Limited ablation on calibration set. [addressed by the rebuttal]

W2: Fair comparison issues. [addressed by the rebuttal]

---
Reviewer abCn

W1: Marginal Practical Gain. [still outstanding]

R1: The authors only re-emphasize efficiency but do not address the concern of "Marginal Practical Gain."

W2: Limited and misleading interpretability. [still outstanding]

R2: The authors do not explain how the HMC’s parameters map to human-interpretable concepts.

W3: Efficacy of RSA. [still outstanding]

R3:  Although with other advantages, it is still true that RSA-selected heads yield lower performance

W4: Robustness and cross-dataset generalization. [addressed by the rebuttal]

**Reviewer Scores:**

Reviewer wJq5: 6 -> 6 (all concerns are addressed, remains positive rating)

Reviewer ypVo: 4 -> 6 (all concerns are addressed, raises the score)

Reviewer TUs8: 4 -> 6 (all concerns are addressed, raises the score)

Reviewer abCn: 4 -> 4 (concerns are still outstanding)

Average score: 5.5

---

### Decision · Program_Chairs · 2026-01-26

Accept (Poster)